# Lidar observations of cirrus cloud properties with CALIPSO from midlatitudes towards high-latitudes

Qiang Li  and Silke Groß

 Deutsches Zentrum für Luft- und Raumfahrt, Institut für Physik der Atmosphäre, D-82234 Oberpfaffenhofen, Germany,

**Correspondence:** Qiang Li (qiang.li@dlr.de)

**Abstract.** Cirrus clouds play a crucial role in the Earth's radiation budget. However, direct observations and model simulations of cirrus at high-latitudes are still sparse. In this study, we present the occurrence rate (OR) and geometrical thickness as well as extinction and particle linear depolarization ratio (PLDR) of cirrus at midlatitudes (35–60°N; 30°W–30°E) and high-latitudes (60–80°N; 30°W–30°E) at temperatures below -38°C using lidar measurements of CALIPSO in the years 2014 and 2018–2021. The results indicate a distinct seasonal cycle in the cirrus occurrence and optical properties. The seasonality in ORs and geometrical thicknesses generally becomes more pronounced with increasing latitude, while the altitude ranges of cirrus decrease with increasing latitude. The extinction coefficients decrease with increasing altitude at both high- and midlatitudes and are, in addition, larger at midlatitudes than at high-latitudes in all seasons. The calculated cirrus optical depths also show larger values at midlatitudes than at high-latitudes, while the differences across latitudes in winter are negligible. The distributions of PLDR in each 5-degree latitude bin show a general decrease with increasing latitude, leading to a remarkable latitudinal difference with larger values at midlatitudes than at high-latitudes. This indicates larger and more irregular ice crystals forming at midlatitudes than at high-latitudes. Finally, we compare the aerosol concentrations at different latitudes acting as ice-nucleating particles (INPs) to trigger heterogeneous freezing, as reported in previous studies. It turns out that aerosols such as mineral dust and soot (including aviation-induced soot) indicate much larger concentrations at midlatitudes than at high-latitudes.

## 1   Introduction

Cirrus clouds are composed entirely of ice crystals with a variety of sizes and shapes and widely occur in the cold upper troposphere (Liou, 1986; Heymsfield et al., 2017; Krämer et al., 2020). Cirrus clouds play a crucial role in modifying the Earth's radiation budget and hydrological cycle (Liou, 1986; Wang et al., 1996; Wylie and Menzel, 1999; Sassen and Benson, 2001; Sassen et al., 2008; Nazaryan et al., 2008). They can efficiently trap outgoing long-wave radiation emitted from the Earth's surface and underlying atmosphere to cause surface warming (greenhouse effect). They also reflect incoming short-wave solar radiation back into space, which results in a cooling effect (albedo effect). The net radiative effect depends on the cloud geometrical, optical, and microphysical properties (e.g. ice crystal size, shape, and orientation in space), which are further determined by the ice formation pathways depending on the surrounding environment (e.g. temperature, humidity, vertical motion, and presence of INPs) (e.g. Fu and Liou, 1993; Zhang et al., 1999; Zerefos et al., 2003; Stephens et al., 2004;

Bailey and Hallett, 2004, 2009; Baran, 2009; Campbell et al., 2016; Krämer et al., 2016; Heymsfield et al., 2017; Marsing et al., 2023). Optically-thin and high cirrus clouds are presumed to have a warming effect because they are nearly transparent to solar incident radiation but still capable to absorb outgoing long-wave radiation (Liou, 1986). Cirrus clouds are typically classified into two types according to the different formation mechanisms and microphysical properties (Krämer et al., 2016; Luebke et al., 2016; Heymsfield et al., 2017). The first type is in situ origin cirrus with ice crystals forming directly from water vapor mainly through homogeneous freezing in cold temperatures (below -38°C) and also through heterogeneous freezing in the presence of INPs. The second type is liquid origin cirrus, which form from glaciation of mixed-phase clouds when they are lifted up into the regions with cold enough temperatures. Satellite observations and climate models reveal that cirrus clouds are one of the most ubiquitous cloud genera with a wide coverage of the Earth's surface, but their fractions display a strong latitude dependence with a maximum of up to 70% over the tropics, approximately 30% over the midlatitudes, and decreasing towards the poles ($\sim 10\%$) (Liou, 1986; Sassen et al., 2008; Nazaryan et al., 2008; Hong and Liu, 2015).

Midlatitude cirrus clouds are of particular importance, not only because they significantly influence the Earth's radiation budget, but also, they interact with the atmospheric dynamics and weather patterns leading to strong uncertainties in their representation in global and regional climate models and numerical weather prediction (NWP) models (Boucher et al., 2013; Kienast-Sjögren et al., 2016; Voigt et al., 2017, 2022; Marquis et al., 2023). They may readily warm or cool the atmosphere during daytime depending on their microphysical properties as well as cloud heights, temperatures, and ice water path (Fu and Liou, 1993; Zhang et al., 1999; Chen et al., 2000; Corti and Peter, 2009). Furthermore, civil aviation takes place mainly in the northern midlatitudes. Aviation emissions lead to the formation of linear contrails and further contrail cirrus which exert a strong influence on the properties of naturally-formed cirrus clouds and contribute a large part of the climate impact of aviation (e.g. Burkhardt and Kärcher, 2011; Tesche et al., 2016; Urbanek et al., 2018; Lee et al., 2021; Li and Groß, 2021, 2022; Groß et al., 2023; *Quaas et al.*, 2024). Since civil aviation over Europe grew strongly before the COVID-19 pandemic and underwent a partial recovery and further increase afterwards in terms of CO2 emission and flight densities, the effects of aviation-induced clouds on our climate are growing significantly. However, there is still a large uncertainty in our understanding of the resulting overall effects (Bock and Burkhardt, 2019; Lee et al., 2021).

In contrast with midlatitudes, cirrus clouds in the Arctic and high-latitude regions are less plentiful and thinner (i.e. lower IWC) (Schiller et al., 2008; Luebke et al., 2013; De La Torre Castro et al., 2023, and references therein). However, they provide a considerable coverage and tend to predominantly warm the atmosphere due to the low elevation of the sun, especially at polar night (Hong and Liu, 2015). It is presumed that there are less efficient INPs available for heterogeneous freezing in the Arctic and high-latitude regions since they are less influenced by anthropogenic pollutants and air traffic than midlatitudes and are lacking in sources of mineral dust and further only small portion of sea salt aerosols can reach cirrus altitudes (Murphy et al., 2019; Wagner et al., 2021). On the other hand, the long-range transport of INPs from midlatitudes (including both natural and anthropogenic sources) to high-latitudes plays an important role in cirrus cloud formation (e.g. *Ratcliffe et al.*, 2024). Especially, smoke aerosols transported across long distances from wildfires in Canada, Alaska, and Siberia frequently impact the Arctic and high-latitude regions by triggering ice crystal formation at lower energy barrier for freezing and by modifying their properties (e.g. Sato et al., 2025; Ansmann et al., 2025a, b). Furthermore, high-latitudes experience more consistent cold

temperatures which leads to more stable and drier air masses compared with midlatitudes. In situ origin cirrus clouds are more prevalent in the high-latitude regions, particularly in polar regions, because of the favourable atmospheric conditions there. Furthermore, deep convection occurs less often at high-latitudes than midlatitudes, which restrains the formation of liquid-origin cirrus clouds. Studies on high-latitude cirrus clouds have been attracting increased attention in recent decades because the near-surface air temperature in the Arctic experiences a faster increase compared to the rest of the world nearly by a factor of four (Rantanen et al., 2022). The enhanced Arctic warming is known as Arctic amplification (AA), which is both a consequence and a driver of feedback processes of climate change (e.g. Serreze et al., 2009). Although the understanding of the formation processes and feedback mechanisms regarding AA has been improving thanks to the intensive studies conducted recently with numerous experimental and theoretical efforts, many uncertainties remain (Holland and Bitz, 2003; Serreze et al., 2009; Serreze and Barry, 2011; England et al., 2021; Rantanen et al., 2022; Wendisch et al., 2019, 2023). Especially, the properties of cirrus cloud in the high-latitude regions and their interaction with AA are still not fully understood. Compared to the midlatitudes where multiple measurements are carried out, high-latitudes are rarely probed due to the lack of sounding instruments. Therefore, a detailed study of cirrus cloud properties and their comparison at different latitudes is highly essential to improve our understanding of cirrus clouds on a global scale.

Many sophisticated ground-based and spaceborne techniques have been developed to observe cirrus clouds. Lidar onboard satellite is the only practical means to probe the atmospheric entities such as clouds on a global scale. Therefore, we use satellite lidar measurements to investigate cirrus cloud properties from midlatitudes towards high-latitudes. In Sect. 2 we will outline the CALIPSO data and methods. Sect. 3 describes our results concerning seasonal variations and long-term trends in cirrus cloud properties and occurrence based on 10-year lidar measurements from March 2010 to February 2020. The dependence of the cirrus cloud properties on the corresponding ambient temperatures as well as the potential impact of aviation are determined and discussed in Sect.4. Our conclusions are finally summarized in Sect.5.

## 2   Data and methods

The CALIPSO satellite equipped with the CALIOP (Cloud-Aerosol Lidar with Orthogonal Polarization) instrument has been providing a comprehensive dataset of the atmosphere observations since June 2006 and advancing our understanding of the atmospheric aerosols and clouds (Winker et al., 2010; Stephens et al., 2018). After 17 years of successful operation, CALIPSO came to an end in August 2023. As the primary payload of CALIPSO, CALIOP is a dual-wavelength elastic backscatter lidar operating at 532 and 1064 nm and has polarization capability at 532 nm (Winker et al., 2007; Hunt et al., 2009). A feature of CALIOP is to continuously observe altitude-resolved profiles of backscatter intensity from the global atmosphere and to identify the vertical structures of cloud and aerosol layers with a high vertical resolution. This is of substantial advantages for relevant studies.

In this study, we use the CALIPSO Version 4.2 Level 2 5-km cloud profile products containing the coded information of particle extinction, backscatter and particle linear depolarization ratio (PLDR) from all the atmospheric entities and additionally the temperatures in cloud derived from the GEOS-5 data. PLDR is a well-defined parameter for evaluating this effect and is

widely used to retrieve information on aerosol profiles and ice particle habits, i.e. particle phase, shape, and orientation (Sassen and Zhu, 2009; Freudenthaler et al., 2009; Tesche et al., 2009; Groß et al., 2012, 2013, 2015). The extinction coefficient refers to a measure to quantify the attenuation of light due to scattering and absorption by ice crystals within cirrus clouds. In contrast with PLDR, extinction is one of the fundamental characterizing bulk properties of cirrus cloud and depends on the particle number concentration. The vertical distribution of extinction is essential for determining the cloud thickness and an improved knowledge of extinction (and thus optical depth) in cirrus clouds would lead to a better estimation of their general albedo effects. In addition, cirrus optical depth is calculated as the integral of extinction along altitudes to investigate the latitudinal difference of cirrus cloud properties. To distinguish cirrus clouds from other features including aerosols as well as from non-cirrus clouds, we have applied the Version 4.2 vertical feature mask (VFM) developed by the CALIPSO team to yield information on feature types and subtypes by decoding the bit-mapped integers recorded in VFM (e.g. Liu et al., 2004, 2009; Hu et al., 2009; Omar et al., 2009; Vaughan et al., 2009; Winker et al., 2009). In addition, to exclude misclassified mix-phased clouds and noise-contaminated signals, we only consider cloud profiles at temperatures below -38°C (= 235 K). For the description of the CALIOP data in more details, readers are referred to Li and Groß (2021) and references therein. The formation heights of cirrus clouds are closely tied to the local tropopause heights which decrease from equator towards poles and cirrus clouds can form at very low altitudes (e.g. as low as 3 km in the extreme cold conditions) in the Arctic and high-latitude regions. Therefore, the cloud profiles above 3 km will be considered for further study, which is different from the previous studies in Li and Groß (2021, 2022).

To study the difference of cirrus cloud occurrence and properties in the high- and midlatitude regions, we focus on the area where the CIRRUS-HL field campaign roughly took place in June-July 2021 (e.g. De La Torre Castro et al., 2023) for a potential cross study in the future, i.e. midlatitudes (35–60°N; 30°W–30°E) and high-latitudes (60–80°N; 30°W–30°E). The map of the research area is shown in Figure S1 (in the supplementary material). We note that the midlatitudes of this research area covers a large fraction of the North Atlantic flight corridor and hence is strongly influenced by aviation emissions and resulting contrails (e.g. Graf et al., 2012; Schumann and Graf, 2013; Voigt et al., 2017; Urbanek et al., 2018; Li and Groß, 2021, 2022; Groß et al., 2023), whereas the high-latitude regions are more pristine. For the current study, the lidar measurements of CALIPSO in 5 years of 2014 and 2018–2021 are analysed, covering pre-COVID years, a heavily COVID-influenced year of 2020, and 2021, a year with relatively mild COVID-19 impacts. The choosing of 2014 is due to the potential cross comparison study between satellite and airborne measurements during the ML-CIRRUS field campaign which took place in March and April, 2014 (Voigt et al., 2017).

## 3 Differences in cirrus cloud properties across latitudes

### 3.1 Occurrence rate and geometrical thickness

Following the analysis in a previous study (Li and Groß, 2021) we removed the data either at temperatures above -38°C or with PLDR values falling into the range below 0.1 and above 0.8 to reject as many as possible unphysical signals with large uncertainties. The pre-analysed observations were further used to infer occurrence rates (ORs) of cirrus clouds as a function of

time (here season), latitude, and altitude. To illustrate the variations of ORs along the latitudes, we calculate the profiles of ORs in each 5-degree latitude bin from 35–80°N. The resulting profiles of ORs in different seasons (spring: March–May; summer: June–August; autumn: September–November; winter: December–February) in different years 2014 and 2018–2021 as well as the composite mean values of the 5 years are shown in Figure 1.

All the profiles of ORs in different seasons show a similar feature that is, in general, the occurrence heights of cirrus clouds decrease with increasing latitude (see Figure S2). In spring, for example, the cirrus cloud ORs derived from all the 5-year observations peak at 10.1 km within the latitude bin of 35–40°N while at 7.1 km within 75–80°N and the altitudes at which the maximum ORs are observed vary at 10.7–7.4 km (9.7–6.8 km, 10.6–7.8 km, 9.8–7.3 km, and 10.2–6.6 km) along the latitude bins in 2014 (in 2018–2021, respectively). The altitude ranges in which cirrus clouds formed as well as the altitudes with the maximum ORs in different seasons can be seen in Figure S1. The seasonal variations in the altitudes with the maximum ORs along the latitudes are discernible, showing the largest values in summer, the smallest in winter, and smaller values in spring than in autumn. The altitude ranges where cirrus clouds form (with a threshold of OR > 1%) generally increase with increasing latitude in both autumn and winter. In spring and summer, however, the ranges only increase with increasing latitude in the midlatitude regions and then decease after the 60°N latitude. Furthermore, there are large variabilities in the altitude ranges of cirrus cloud in different years, especially for those in spring at high-latitudes. A distinct seasonal variation is visible, showing the values of cirrus altitude range varying from 5.5 km (found within 35–40°N in 2021) up to 8 km (found within 55–60°N in 2020) in winter and from 2.7 km to 4.8 km in summer. In spring and autumn, however, the values fall within the range from ∼4.8 to 6.7 km. The details of altitude ranges of cirrus cloud formation (with OR > 1.0%) calculated from all the 5-year observations in different seasons are summarized in Table 1 and a schematic representation can be seen in Figure S1. Furthermore, the ORs of cirrus cloud themselves show a clear seasonal variation with the largest values in winter, the smallest in summer, and larger values in autumn than in spring. These findings are consistent with the results in previous studies (e.g. Sassen et al., 2008; Li and Groß, 2022). The ORs in 2014 are larger than the composite mean values of OR derived from all 5-year observations along the latitudes for all seasons, especially at latitudes from 50°N polewards. These changes in the ORs of cirrus cloud were also indicated in Li and Groß (2022) for a smaller research area at midlatitudes. This is likely due to the fact that air temperatures increased in the past years and even faster at high-latitudes due to AA (e.g. Li and Groß, 2022; Rantanen et al., 2022). To clarify the impacts of meteorological conditions on cirrus cloud formation, we further compare the background temperatures and relative humidity with respect to ice (RHi) directly derived from global ERA-5 re-analysis data (Hersbach et al., 2020) covering the research area in this study (not shown here). Indeed, the temperatures in 2014 are generally lower than in more recent years, especially in the summer months. The RHi values, however, are higher in 2014 than in other years, especially at midlatitudes. Therefore, the meteorological conditions in 2014 are more favourable for ice crystal formation along the latitudes than in the years 2018–2021.

In order to compare the morphologies of cirrus clouds at high-latitudes (HL) and midlatitudes (ML), we divide the data into a subset for HL (60–80°N; 30°W–30°E) and ML (35–60°N; 30°W–30°E) following the commonly-used definitions in previous studies (e.g. De La Torre Castro et al., 2023), although the partitioning of latitude ranges is somehow arbitrary due to lacking of universally-accepted definition of latitude ranges. The composite mean values of cirrus ORs in different seasons calculated

from all the 5-year lidar observations in years of 2014 and 2018–2021 are shown in Figure 2. Apparently, cirrus clouds at HL formed mostly at lower altitudes than those at ML due to the fact that favourable atmospheric conditions involving temperature and humidity for ice crystal formation can extend to lower altitudes at HL. The differences between the altitudes at which the maximum ORs are observed at HL and ML show a distinct seasonal dependence. For example, the height difference of maximum ORs is 1.8 km in winter but is only 1.1 km in summer. The heights of cirrus cloud formation typically range from near 7.0 to 12.0 km ( 5.0–12.5 km) for HL and 7.8 to 13.2 km ( 6.0–13.5 km) for ML in summer (in other seasons). The cloud tops of cirrus are very similar in different seasons (roughly 12.5 km at HL and 13.5 km at ML) whereas the cloud bottoms are much higher in summer than in other seasons. In addition, the values of OR show much larger variabilities at HL than at ML, which is related to the larger variabilities in humidity at HL than at ML and is consistent with a recent model study showing larger INP effects on cirrus at higher latitudes (Beer et al., 2024). The ORs of cirrus cloud at HL can get as large as 15.03% at 7.9 km in winter but only 5.2% at 9.1 km in summer. At ML, however, the maximum ORs are within the range from 7.1% to 10.8%.

We next analyse the geometrical thickness of cirrus cloud, which is defined as the vertical distribution of cirrus clouds at different latitudes (e.g. Li and Groß, 2021). We first group cirrus clouds with geometrical thicknesses larger than 100 m, 300 m, 1 km, and 2 km, respectively, based on a single profile of observations. The relative probabilities of the occurrence in different seasons are then calculated as the ratio of the number of profiles with cirrus cloud layers to the total number of observed profiles. The resulting frequency of occurrence according to different cloud thickness thresholds are shown in Figure 3. First, there is a distinct seasonal cycle in the geometrical thickness of cirrus clouds with the largest values in winter and the smallest in summer for both HL and ML and for all the thickness thresholds. In addition, the comparisons of cloud thickness in different seasons show that the thickest cirrus clouds formed in winter and the thinnest ones in summer. The cloud thickness at ML are very close to each other in spring and autumn whereas at HL they are thinner in spring than in autumn. Comparisons in different seasons show that the cloud thicknesses in spring and summer (in autumn and winter) are thicker (thinner) at ML than at HL (Sassen et al., 2008; Nazaryan et al., 2008; Stubenrauch et al., 2013; Gasparini et al., 2018). Therefore, there are larger variabilities in cloud thickness at HL than ML in all seasons, which is presumed to be caused by the larger variabilities in the humidity (RHi) and lower air temperatures at HL than ML (not shown here) as well as the larger INP effects on cirrus at higher latitudes (e.g. Beer et al., 2024). The results in different years show remarkable reductions in cirrus thickness only at ML in spring and summer of 2020–2021 and in winter of 2021, which is very well following the timeline of the aviation reduction during the COVID-19 pandemic (e.g. Li and Groß, 2021). The changes in cloud thickness induced by the aviation reduction are presumed to be smaller at HL than ML since the HL regions are much more pristine and are subjected to the larger variabilities of cirrus clouds at HL. In addition, we also compare the monthly geometrical thicknesses of cirrus clouds at ML and HL (not shown here). The results show that there are stronger seasonal variations in the geometrical thickness of cirrus at HL than ML. For the cases with the definition of cloud thickness larger than 0.3 km, for example, cirrus clouds at HL reached the maximum frequency of 39.3% in January and the minimum frequency of 13.5% in June whereas at ML they reached the extremes of 32.1% in December and 15.5% in July, respectively. Further, the values are very similar at HL and ML

**Table 1.** Altitude range of cirrus clouds calculated from the composite 5-year observations in different seasons with ORs larger than 1.0%.

| Latitude (°N) | 35–40 | 40–45 | 45–50 | 50–55 | 55–60 | 60–65 | 65–70 | 70–75 | 75–80 |
|---|---|---|---|---|---|---|---|---|---|
| MAM (km) | 5.40 | 5.61 | 5.52 | 5.70 | 6.06 | 6.06 | 6.03 | 5.94 | 5.79 |
| JJA (km) | 3.18 | 4.08 | 4.38 | 4.50 | 4.50 | 4.35 | 3.99 | 3.48 | 3.36 |
| SON (km) | 5.16 | 5.55 | 5.64 | 5.82 | 5.97 | 5.94 | 6.12 | 6.21 | 6.39 |
| DJF (km) | 5.91 | 6.57 | 6.99 | 7.20 | 7.05 | 7.20 | 7.38 | 7.23 | 7.14 |

in April and August and become smaller at HL than ML in the warmer months (May–July) and larger at HL than ML in the colder months (September–March).

## 3.2 Extinction coefficient and cloud optical depth

CALIOP is a backscatter lidar and actually measures attenuated backscatter profiles containing information on backscatter and extinction. Extinction profiles in CALIPSO Version 4 data are derived using a constrained Klett-Fernald inversion of the lidar equation with a temperature-dependent lidar ratio applied for cirrus clouds, i.e. the initial lidar ratio from ∼35 to 21 sr with the layer centroid temperature from 0 to -90 °C (Young and Vaughan, 20009; Young et al., 2013, 2016, 2018). For dense or opaque layers, however, Version 4 implements an initial lidar ratio from measurements rather than assuming a constant value (Young et al., 2018). The assumption of lidar ratio is combined with a multiple-scattering correction factor to account for the effect of photons scattering more than once. In Version 4, the multiple-scattering factor used in cirrus extinction retrievals is parameterized as a function of layer centroid temperature (Garnier et al., 2015). The lidar ratio of cirrus can vary significantly depending on ice crystal size, habit, and ice water content. The assumption of the lidar ratio introduces a significant amount of uncertainty into extinction retrievals, which dominates the total uncertainty. Table 2 summarizes the relative uncertainty contributions to the extinction retrievals of cirrus cloud from different sources reported in literatures, although the exact values depend on altitude, optical depth, and day or night conditions. In total, the relative uncertainty for cirrus clouds is estimated to be ∼20-25% in Version 4 for typical cirrus extinction profiles that is well improved compared with the previous version (Young et al., 2018).

Due to the temperature-dependent IWC-extinction relationship, IWC is calculated as a parameterized function of the retrieved extinction coefficients within ice crystals of cirrus clouds involving temperature and effective diameter of ice crystals and is stored in the Level 2 cloud profile product (Heymsfield et al., 2005; Mioche et al., 2010; Heymsfield et al., 2014). In the current study, we only compare the profiles and distributions of extinction coefficients of ice crystals at HL and ML.

The resulting vertical profiles of cirrus extinction medians in spring (MAM: March–May) as well as the 25th and 75th percentiles from all 5 years combined are shown in Figure 4 (left panel). It is mentioned in Section 2 that the differences in cirrus cloud properties induced by the COVID-19 pandemic are presumed to be the largest. The results of cirrus extinction in spring are hence first compared. In general, they reveal a decrease with increasing altitude for all the cases in different years and in different latitude ranges (i.e. at HL and ML, respectively), indicating smaller and/or fewer ice crystals at higher altitudes

for both latitude domains. Besides the differences in the altitudes containing cirrus cloud formation (with ORs > 1.0%) at different latitudes (with nearly 1.5-km discrepancy; See more descriptions in details in Subsection 3.1), cirrus clouds probed at HL are characterized with smaller extinction coefficients than those at ML (e.g. Gasparini et al., 2018). The latitudinal dependence is closely linked to the difference in the background meteorological conditions, dynamics, and availability of INPs. Cirrus clouds at high latitudes predominantly form in situ in colder and drier air masses, resulting in lower ice crystal number concentrations, smaller particle sizes, and hence optically thinner cirrus; while at midlatitudes, in contrast, cirrus clouds originate from the freezing of supercooled liquid clouds associated with stronger frontal systems or convective outflow, which favours the formation of thicker cirrus layers (Luebke et al., 2016; Krämer et al., 2016; Gasparini et al., 2018). Further, the differences between cirrus extinction coefficients at HL and ML become smaller with increasing altitudes from 0.14 to 0.03 km$^{-1}$ derived from the composite values of 5-year observations.

To compare the cirrus extinction coefficients at different latitude domains, we show box plots in the right panel of Figure 4 for the extinction coefficient distributions in spring in each individual year and all 5 years combined. At ML, the year-to-year variations in extinction coefficients are discernible with a slight increase from 2014 to 2019 with the medians from 0.175 to 0.205 km$^{-1}$ and with notable reductions in 2020 and 2021 with the medians of 0.169 and 0.160 km$^{-1}$, respectively, which might be partially related to the aviation reduction during the COVID-19 period. However, the influences of variable meteorological conditions cannot be ruled out (e.g. Schumann et al., 2021a; *Quaas et al.*, 2021; Li and Groß, 2021). At HL, however, extinction coefficients are comparable in different years with an exception of a small enhancement in 2019 with medians from 0.131 to 0.146 km$^{-1}$. The comparison of extinction coefficient distributions across latitudes shows a clear consistence in different years, i.e. the extinction coefficients are larger at ML than at HL. From the composite values in 5 years, the extinction coefficients are within the range of 0.012 to 1.284 km$^{-1}$ (considering the 5% to 95% percentile of the data set) at ML and from 0.010 to 1.005 km$^{-1}$ at HL, respectively. The distributions of extinction coefficients in other seasons are also shown in Figure S3. From Figure S3 (Panels a, b, d, e), the seasonality in extinction coefficient distributions is noticeable, showing the largest extinction coefficients in winter and the smallest in summer. In addition, the latitudinal comparison shows a stronger seasonal variation at HL than at ML. The derived extinction from the current study shows slightly larger values than the results of Ansmann et al. (2025a) who carried out cirrus cloud observations with lidar and radar aboard the German ice breaker *Polarstern* at latitudes > 85°N. The comparison provides a further support that cirrus cloud extinction shows a decrease with increasing latitudes. Considering the impact of polar day and night, we also show the results only from the day-time measurements in summer (Panel S3c) and only from the night-time measurements in winter (Panel S3f) for a fair comparison between HL and ML. In general, the diurnal variations of extinction coefficient are notable, which show larger values during day time than night time for both seasons (the same diurnal variations are also seen in spring and autumn). And further, the diurnal variations are stronger at ML than at HL, which backs up the necessity of considering the impacts of polar day and night. Furthermore, the year-to-year variations are stronger in autumn and winter than in spring and summer for both latitude ranges. In particular, extinction coefficients during the COVID-19 pandemic are in general smaller than those in the other non-COVID years at ML, whereas at HL, the reductions in extinction coefficient cannot be clearly recognized.

**Table 2.** Relative uncertainty contributions to CALIPSO cirrus extinction retrievals reported in literatures.

| Source of uncertainty | Relative uncertainty (%) | References |
| --- | --- | --- |
| Lidar ratio | ∼10-15% | Young et al. (2013, 2018) |
| Daytime SNR loss | ∼10-15% | Young et al. (2013) |
| Multiple scattering effects | ∼5-10% | Josset et al. (2012); Garnier et al. (2015); Young et al. (2018) |
| Photon noise | ∼5-10% | Young et al. (2013) |
| Undetected layers or misclassification | ∼5-10% | Thorsen and Fu (2015) |
| Reference altitude selection | ∼5% | Young et al. (2013) |
| Calibration errors | ∼5% | Vaughan et al. (2019) |

Cirrus optical depth is a good measure to quantify the overall impact of cirrus clouds on the absorption and scattering of light passing through the cloud. Previous studies indicated that cirrus is the only cloud genus capable of inducing either cooling or heating during daytime at the top of the atmosphere (TOA) and the radiative balance greatly depends on their optical depth (Lolli et al., 2017; Campbell et al., 2016, 2021). Although studies on the cloud optical depth of cirrus have been carried out intensively, investigations of its geometrical and optical properties over the Arctic are still rare. The optical depths of cirrus clouds are calculated as the integral of the extinction over the altitudes of the vertical distributions of cirrus. The resulting histograms of cirrus optical depth, for example, in spring in different years of 2014 and 2018–2021 are shown in Figure 5. First, the histograms of cirrus optical depths exhibit a left-skewed distribution with a long tail extending to smaller values other than a normal distribution in the logarithmic scales. Cirrus clouds at ML are characterized with slightly larger cirrus optical depth than at HL. That is, the values derived from the composite data of the 5 years are within the ranges from ∼0.005 to 1.258 (again from 5% to 95% percentile of the data set) for cirrus clouds at HL and from ∼0.004 to 1.373 at ML. The here-derived optical depth of cirrus compared with the results of Ansmann et al. (2025a) shows slightly larger medians and larger upper thresholds, although both datasets show a comparable left-skewed distribution. We also note that lower thresholds of the distributions are smaller in 2020 and 2021 than in other years for both HL and ML. Furthermore, the corresponding medians for different years consistently show larger values at ML than HL with a difference of 0.015 from the composite data of the 5 years. Please note the deviation of medians in 2020 from other years might be correlated with the late phase of anomalies in the Arctic polar vortex in 2019/2020 (e.g. Lawrence et al., 2020; Huang et al., 2021) which could increase the optical depth of cirrus by lifting moist air into higher and colder altitudes to enhance the formation of smaller ice crystals. Indeed, the meteorological conditions derived from the ERA5-reanalysis data reveal lower temperatures and enhanced RHi in the spring months in 2020 than in other years (not shown here). In addition, seasonal variations in cirrus optical depth are also noticeable with the largest values in winter and the smallest in summer (see Figures S4-S6 for the histograms of cirrus optical depth in other seasons). The differences in cirrus optical depth across latitudes (i.e. larger values at ML than at HL) are more notable in spring and summer than in autumn. In winter, however, the values are very close between different latitudes.

### 3.3 Particle linear depolarization ratio

As mentioned above the CALIOP lidar is polarization-sensitive at 532 nm and is able to independently measure two orthogonal polarization components which are polarized parallel and perpendicular to the polarization plane of the transmitted beam. Thus, the Level 2 cloud profile products also contain the information of particle linear depolarization ratio (PLDR). We next turn to compare the distributions of PLDR in different years across latitudes.

In Figure 6, we present the distributions of cirrus PLDR in box plots for each 5-degree latitude bin in spring (Mar–May) derived from both day- and night-time observations from years of 2014 and 2018–2021 as well as from the composite values of 5 years. For all the cases, the mean values of PLDR are larger than the corresponding medians, implying that the PLDR values follow a positively-skewed distribution (e.g. Li and Groß, 2021). The year-to-year variabilities in PLDR show larger values in the pre-COVID years (2014, 2018, and 2019) than in 2020 and 2021 under the aviation reduction during the COVID-19 pandemic at midlatitudes (ML) (e.g. Li and Groß, 2021). At high-latitudes (HL), however, the PLDR values decrease with increasing latitude only in 2014 and 2018 and become nearly constant in 2019 and 2021. The case in 2020 is distinctive from other years, showing that PLDR tend to slightly increase with increasing latitude (only at latitudes larger than 60°N). This leads to the comparable values in the PLDR distribution at ML and HL because of the reduction in PLDR at ML caused by the aviation reduction due to the COVID-19 restriction (Li and Groß, 2021) and additionally because of enhancement in PLDR at HL in connection with the unusual strong stratospheric polar vortex (e.g. Lawrence et al., 2020; Huang et al., 2021). The corresponding results in other seasons are indicated in Figures S7-S9 in the supplementary material, which present similar characteristics as the PLDR analysis in spring. Besides the year-to-year variabilities, the composite values of cirrus PLDR of the 5 years show a clear decrease with increasing latitude for all the seasons. A significance test for the derived decreasing trend in PLDR with latitude is carried out applying the Mann-Kendall (MK) test (Mann, 1945; Kendall, 1975). The $p$ values are calculated from the MK test, which measure the probability of rejecting the null hypothesis, i.e., the data are independently distributed with no trend. The exercises have been done to the values of PLDR in different seasons from different years as well as from all 5 years combined. It is striking that the results of the MK test from the combined 5 years of data in different seasons show that there are significantly decreasing trends in PLDR with increasing latitude with the $p$ values much smaller than 0.001. The same results are derived from the data in different seasons from each individual year. Only exception is found in the case in Spring (MAM) 2020 that the latitudinal gradient in PLDR is smoothed out by the combined influence of the reduced PLDR due to aviation reduction at ML and the enhanced PLDR at HL due to strong polar vortex. As a summary, we provide the quartiles (Q1, Q2, and Q3 = 25th, 50th, and 75th percentiles, respectively) of all the PLDR values from the composite 5-year observations in spring in Table 3.

We next compare the PLDR distributions of cirrus clouds in different seasons at HL and ML, respectively, in more detail in Figure 7. Besides the seasonality, PLDR are generally larger at ML than at HL, especially in spring and autumn. The only exception is spring 2020 with slightly larger PLDR at HL than at ML, as mentioned before. Similarly, the enhancements in PLDR at HL are also seen in autumn 2019 and winter 2019/2020 compared to the corresponding seasons in other years. The reasons for the enhanced PLDR at HL are presumed to relate to lower temperatures as well as higher aerosol loading

**Table 3.** The quartiles of all the PLDR values from the composite 5-year observations in spring (Mar–May) within the altitudes 6–12 km.

| Latitude (°N) | 35–40 | 40–45 | 45–50 | 50–55 | 55–60 | 60–65 | 65–70 | 70–75 | 75–80 |
|---|---|---|---|---|---|---|---|---|---|
| Q1 | 0.266 | 0.261 | 0.252 | 0.247 | 0.244 | 0.240 | 0.242 | 0.236 | 0.237 |
| Q2 | 0.365 | 0.361 | 0.353 | 0.347 | 0.342 | 0.335 | 0.339 | 0.332 | 0.336 |
| Q3 | 0.472 | 0.471 | 0.468 | 0.463 | 0.458 | 0.451 | 0.456 | 0.452 | 0.458 |

(including soot and smoke) caused by the long-lasting strong stratospheric polar vortex during this period (e.g. Manney et al., 2022; Ansmann et al., 2023). Besides the reductions in PLDR at ML during the COVID-19 restriction, the values of PLDR at ML are relatively stable, showing small year-to-year variabilities. In contrast, the year-to-year variabilities in PLDR at HL are much larger in all seasons. The impacts of the COVID-19 on the cirrus cloud properties at HL are difficult to identify since the potential COVID-induced changes in PLDR are too small and are obscured by the large year-to-year variabilities. We note that the lidar measurements at HL in summer are mostly derived during daytime (polar day) and, in contrast, the measurements at HL in winter are mostly at night (polar night). Previous studies reveal that aviation exerts a stronger influence on cirrus cloud properties in the daytime than at night (e.g. Graf et al., 2012; Schumann and Graf, 2013; Li and Groß, 2022). Therefore, we also derive the results from only the daytime measurements in the summer months and from only the night-time measurements in winter at ML and HL for a fair comparison. The resulting distributions of cirrus PLDR are shown in the Panels c and f of Figure 7. Diurnal variations in PLDR are remarkable, showing larger values during daytime than at night (e.g. Sassen and Zhu, 2009; Li and Groß, 2022). For the comparison during daytime in summer, the difference of PLDR between ML and HL becomes much larger than the results derived from both day- and night-time observations. However, the night-time measurements in winter (Panel f) show the opposite, namely the differences in PLDR across latitudes become smaller compared with the results from both day- and night-time observations. The findings are backed up by the fact that aviation densities are larger during daytime than at night (e.g. Graf et al., 2012; Schumann and Graf, 2013).

## 4 Discussion

### 4.1 Implications of previous studies

Although the current understanding of cirrus cloud properties has been improved with theoretical and experimental efforts over the last decades, there are still many remaining uncertainties, especially the impacts of aviation emissions. To our knowledge, Urbanek et al. (2018) is one of the first studies to show the potential effect of aviation emissions on cirrus cloud properties. With airborne lidar measurements through specific clouds, they indicated that enhanced heterogeneous freezing on aviation exhaust particles is responsible for the resulting lower supersaturation and larger particle linear depolarization ratios (PLDR) (Urbanek et al., 2018). The findings were further backed up by the satellite measurements with CALIPSO. Namely, the aviation-induced changes in cirrus cloud properties can be characterized by the distributions of cirrus PLDR either during a specific period with

a strong aviation reduction due to the COVID-19 pandemic or during a 10-year term with a slight increase in aviation emissions (Li and Groß, 2021, 2022).

## 4.2 Ambient temperatures in the high- and midlatitude regions

Our analysis above shows that, besides seasonality, there are significant deviations in the occurrence rates (OR) and optical properties of cirrus clouds forming at different latitudes. It has been reported that temperatures govern ice crystal formation pathways and further affect crystal habit and shape (Koop et al., 2000; Bailey and Hallett, 2004, 2009; Krämer et al., 2016, 2020). Furthermore, the optical properties of cirrus cloud (e.g., PLDR) significantly depend on the ambient temperatures (Urbanek et al., 2018; Li and Groß, 2021). We thus examine the relationship between cirrus cloud properties and the ambient temperatures inside cirrus clouds.

Previous studies reveal that cirrus cloud formation and morphologies as well as the high degree of variability in their microphysical properties strongly depend on the background meteorological conditions, especially on temperatures (Urbanek et al., 2018; Li and Groß, 2021). A direct comparison between PLDR and the ambient temperatures showed that the PLDR values decrease with rising temperatures at temperatures below -50°C and become stable at warmer temperatures higher than -50°C (Li and Groß, 2021). These findings help to interpret the potential causes for the latitudinal differences of cirrus cloud properties. The datasets of temperature used here are derived from the GEOS-5 (Goddard Earth Observing System, version 5) model data product provided to CALIPSO by the Global Modelling and Assimilation Office (GMAO) data assimilation system.

Figure 8 shows the histograms of temperature inside cirrus clouds in spring of different years at HL and ML, respectively. The distributions show a clear consistence in different years at both latitude ranges. The cirrus cloud formation at ML took place with the maximum occurrence at temperature of -38°C, the temperature mask we chose to filter out aerosols and non-cirrus clouds from the dataset, and with a "shoulder" as temperature reached ∼-60°C. Cirrus clouds at HL, however, occurred at temperatures following a quasi-lognormal distribution with peaks at ∼-45°C with a cutoff at -38°C due to the temperature filtering. For both cases, the histograms of temperatures inside cirrus clouds can be characterized by a left-skewed distribution with a long tail extending to lower temperatures. The comparisons between different latitudes show larger medians at ML than HL by nearly ∼1°C in 2014, 2018, and 2019, and nearly identical medians in 2021. The case in 2020 shows smaller medians at ML than HL by 1.5°C and the histograms at HL with a longer tail extending to -76.2°C, which is related to the lower temperatures caused by the strong stratospheric polar vortex lasting from autumn 2019 to spring 2020. Regarding all 5-year measurements, the values are very close across latitudes in terms of the medians and statistical distributions. For the 5-year measurements in other seasons (see Figures S10-S12), the medians of temperatures inside cirrus clouds are larger at ML than HL by 3.1 and 1.6°C in summer and autumn, respectively, but smaller at ML than HL by 1.6°C in winter. Based on these comparisons, we may conclude that temperatures inside cirrus clouds can likely be excluded as the cause for the difference in the cirrus extinction and PLDR at different latitudes as shown in the present study.

## 4.3 Aerosol loading and aviation emissions in the high- and midlatitude regions

As mentioned above the research area in this study covers a large portion of the northern hemisphere spanning from the northern Atlantic to the European mainland and extending into the Arctic. It is characterized by a large diversity of geography, climate, and environment including a wide range of aerosol types. The presence of aerosol particles in the atmosphere can catalyse the formation of ice crystals by aerosols acting as ice-nucleating particles (INPs) which induce heterogeneous freezing at sub-zero temperatures > -38°C and lower supersaturations with respect to ice compared to homogeneous freezing (Koop et al., 2000;

Hoose and Möhler, 2012). With the airborne lidar measurements of WALES, Urbanek et al. (2018) indicated that the increased heterogeneous freezing caused by aviation emissions can be responsible for the enhanced PLDR of cirrus clouds. The finding is further supported by the microphysical properties of the high-PLDR-mode cirrus clouds, which exhibit larger effective ice particles and lower number concentrations confirming the effects of enhanced heterogeneous freezing (Groß et al., 2023).

The tiny aerosol particles suspended in air vary widely in type and composition, which leads to significant variations in

the microphysics of ice clouds and further influences their optical and radiative properties and climate effects (Boucher et al., 2013). However, INP concentrations are typically very low in the atmosphere, with only a tiny fraction of $10^{-3}$ to $10^{-5}$ of the ambient aerosols acting as INPs (Rogers et al., 1998) and even a lower fraction in marine regions (Rosinski and Morgan, 1988). INPs can be natural or anthropogenic particles, originating from dust storms, volcanic eruptions, sea spray, biogenic emissions, and anthropogenic activities (including industrial emissions, vehicle exhaust, and urban pollution) (Hoose and Möh-

ler, 2012; Kulkarni et al., 2016; Kärcher, 2017; Kanji et al., 2017; Gao et al., 2022; Beer et al., 2022, 2024). The distribution of aerosol particles in the high-latitude and midlatitude regions can vary significantly due to the differences in their sources. At midlatitudes, aerosol types are predominantly sulphate, black carbon and organic carbon with anthropogenic sources as well as mineral dust and sea salt from natural emissions; whereas at high-latitudes, in contrast, the overall aerosol concentrations are lower with more natural aerosols including sea salt (mostly at lower levels), biogenic particles, and wildfire smoke. Neverthe-

less, the aerosol types and compositions across latitudes can influence each other due to transport mechanisms and atmospheric circulation patterns. According to recent model simulations (Beer et al., 2022, 2024), mineral dust and black carbon that are commonly considered as the main INP types in global models show a distinct north-south gradient with larger concentrations at midlatitudes than high-latitudes.

Among the abundant sources of aerosol, wildfires emit large amounts of smoke aerosols embedding a mixture of potential

INPs (including soot, organic matter, mineral dust, and etc) for ice crystal formation in the upper troposphere and lower stratosphere (UTLS). (Khaykin et al., 2018; Ansmann et al., 2018; Mamouri et al., 2023). Wildfire smoke provides surfaces for triggering heterogeneous ice nucleation at warmer temperatures and lower water vapor supersaturation which suppresses homogeneous nucleation of solution droplets (Gierens, 2003; Kärcher et al., 2022). This leads to the formation of fewer, larger, and more irregular ice crystals, exhibiting enhanced PLDR and likely reduced extinction and optical depth (Veselovskii et al.,

2022; Mamouri et al., 2023; Ansmann et al., 2025a). Furthermore, modelling studies suggest that smoke particles may also increase the sedimentation rates of ice crystals and hence lead to cirrus cloud thinning (Ansmann et al., 2025b). However, there is no precise study on this topic that is well constrained in observations. Due to different backgrounds in the two regions of

high- and midlatitude, the contribution of smoke influence on cirrus clouds can be very different. Smoke transported into the very cold and stable UTLS at high-latitudes can suspend very long, providing a persistent INP reservoir. Smoke at midlatitudes is normally episodic and mixed often with other aerosols and airmasses. Hence, additional smoke aerosols can reduce the latitudinal gradient of aerosol impact on cirrus clouds in the two latitude domains.

In addition, soot particles from aviation emissions are presumed to play a determinate role in the formation of contrails and contrail cirrus, since aviation exhaust particles are mainly emitted at cirrus cloud altitudes leading to aviation soot number concentrations at flight altitude often exceeding those of other aerosols (Righi et al., 2021). While soot particles from modern aircraft engines may not be highly efficient INPs for direct ice crystal formation at cirrus temperatures (Testa et al., 2024a, b; Yu et al., 2024), they still play a crucial role in the formation of contrails and further evolving into contrail cirrus (Chauvigné et al., 2018), especially at midlatitudes with heavy air traffic. However, cirrus cloud formation is very complex depending on atmospheric conditions of temperature, humidity, vertical motion and other coexistent aerosols. Furthermore, soot particles may undergo multiple cloud cycles during their residence time in the atmosphere, which can improve their porosity and surface wettability and thus change their ability for pore condensation and freezing (PCF) (e.g. Mahrt et al., 2018; Gao et al., 2022). In addition, previous studies indicated that soot particles involved in ice crystal formation may lead to more irregular ice crystal shapes in cirrus cloud compared to ice crystals nucleated homogeneously or on typical mineral dust INPs depending on their surface features and chemical compositions (Kärcher and Lohmann, 2003; Hoose and Möhler, 2012), which is responsible for the enhanced PLDR of cirrus clouds (Urbanek et al., 2018; Li and Groß, 2021, 2022). Due to various factors including population distribution, social and economic activities, and geographical constrains, aviation densities at midlatitudes are much larger than at high-latitudes (by a factor of more than 10 times) (Stettler et al., 2013; Teoh et al., 2024).

## 5   Conclusions

In the last decades, the Arctic has been warming at a faster rate compared to the global average, which is known as Arctic amplification (AA). The thin cirrus clouds in the Arctic that exert a substantial positive forcing on the surface temperatures can trigger AA and further accelerate the processes of warming, especially in polar winter. Compared to the intensive studies of cirrus clouds in the tropics and midlatitude regions, however, direct measurements and model simulations at high-latitudes are still very rare. In the current paper, we present an analysis of cirrus clouds from midlatitudes (ML, 35–60°N; 30°W–30°E) towards high-latitudes (HL, 60–80°N; 30°W–30°E) with lidar measurements of CALIPSO in the years 2014 and 2018–2021. The derived occurrence and properties of cirrus clouds have been compared across latitudes.

The resulting profiles of occurrence rates (ORs) of cirrus clouds in different seasons follow a distinct seasonal cycle with the largest values in winter, the smallest in summer, and larger values in autumn than in spring. The seasonal cycles in ORs at HL are much stronger than at ML with nearly 10% difference of ORs at HL compared to only 3.7% at ML. Furthermore, the heights of cirrus cloud formation show a clear decrease with latitudes. Depending on the meteorological conditions for ice crystal formation, cirrus clouds in summer only appear at higher altitudes compared to other seasons, which results in similar cloud top heights in all seasons but much higher cloud bottom heights in summer than in other seasons. The geometrical

thicknesses of cirrus clouds also show a distinct seasonal cycle with the largest values in winter and the smallest in summer at both HL and ML and for all applied thickness thresholds. The seasonality at HL is stronger than at ML. Furthermore, the cloud thicknesses are also compared across latitudes and show larger (smaller) values at ML in spring and summer (in autumn and winter) than at HL.

We next compare the extinction coefficients of cirrus clouds and their vertical profiles show a decrease with increasing altitude in different years within both HL and ML regions. Besides the 1.5-km differences in the altitudes of cirrus cloud formation at different latitudes in spring, cirrus clouds observed at HL are characterized with smaller extinctions than those at ML. The differences in extinctions at different latitudes may be connected to the nucleation processes of ice crystals depending on the dynamical and thermodynamic conditions and the availability of INPs. Notably, the year-to-year variations in extinctions
show a slight increase from 2014 to 2019 and a noticeable reduction in 2020 and 2021 at ML while staying nearly identical in different years at HL. Further, they are stronger in autumn and winter than in other seasons for both HL and ML. In addition, the seasonal variations in extinction show the largest values in winter and the smallest in summer, which are stronger at HL than at ML. We notice that cirrus cloud extinctions during the COVID-19 pandemic (starting from spring 2020) are in general smaller than those in the pre-COVID years at ML, whereas there are no clear reductions in extinctions being recognized at HL.
The cloud optical depths of cirrus show, in general, the same seasonality as extinctions with larger values at ML than at HL except for the winter months.

Particle linear depolarization ratios (PLDR) decrease with increasing latitude from all the observations of 5 years combined. However, the characteristics of PLDR distributions vary in different years. The year-to-year variabilities in PLDR show larger values in the pre-COVID years (2014, 2018, and 2019) than in 2020 and 2021 that are likely influenced by aviation reductions
during the COVID-19 pandemic at ML. At HL, however, PLDR show no clear reductions during the same period of COVID-19. This is likely due to the fact that weak aviation effects on cirrus clouds at HL have immerged into the large year-to-year variabilities caused by other conditions.

To study the potential causes for the latitudinal differences in cirrus cloud properties, we first determine the distributions of temperatures inside cirrus clouds at different latitudes. Their histograms show a similar distribution in different years at both
HL and ML, following a left-skewed distribution with a long tail extending to lower values. At ML, cirrus clouds mainly formed within the temperature range from -60° to -38° with the maximum occurrence at -38°. At HL, however, the temperatures inside cirrus follow a quasi-lognormal distribution with a cutoff at -38°, peaking at ∼-45°. In spring, temperatures inside cirrus are characterized with larger medians at ML than at HL in 2014, 2018, and 2019, smaller medians at ML than at HL in 2020, and nearly identical medians in 2021. In other seasons, however, the medians of temperatures inside cirrus across latitudes
vary slightly with differences up to ∼3°C determined from the composite values of 5-year measurements. Due to the weak latitudinal dependency, temperatures within cirrus clouds are likely not the cause of the latitudinal differences in cirrus cloud properties shown here.

Finally, we compare cirrus cloud properties in regions of different aerosol concentrations as reported in previous studies, which can act as ice-nucleating particles (INPs) to form ice crystals via heterogeneous freezing. The distribution of aerosol
particles across latitudes can vary significantly due to the different sources of these particles, i.e. with higher aerosol con-

centrations from more anthropogenic sources at ML and lower concentrations with more natural aerosols at HL. Although transport mechanisms and atmospheric processing may influence the aerosol compositions and concentrations at different latitudes, there is a distinct north–south gradient in the number concentrations of dust and soot with larger values at ML than at HL. In addition, aviation-induced soot particles that are more significant at ML than at HL are emitted directly into the cirrus regime and could often show larger concentrations compared to other aerosols at aviation cruising altitudes. Although soot particles from modern aircraft engines may not be efficient INPs (Testa et al., 2024a, b; Yu et al., 2024), they can act as condensation nuclei for forming tiny water droplets, especially, as they aggregate and mix with other substances. These processes will influence ice cloud formation indirectly by competing with other aerosols for available water vapor and suppressing homogeneous nucleation. Furthermore, soot particles play a crucial role in the formation of contrails and contrail-cirrus. They may undergo multiple cloud cycles during their residence time in the atmosphere and thus improve their ice-nucleating ability via PCF. Notably, ice crystals forming through heterogeneous freezing are characterized by larger sizes and more irregular shapes. In turn we hypothesize that the differences between heterogeneous and homogeneous freezing depending on latitudes may be responsible for the observed latitudinal dependency of cirrus cloud properties. This work highlights the differences in the optical properties of cirrus cloud across latitudes that is crucial for improving global climate models and understanding the important role cirrus clouds play in Earth's radiation budget and climate feedback. However, CALIOP suffers in several limitations such as low signal-to-noise ratio, daytime limitation, low sampling resolution, etc. In the future work, we will make use of the lidar measurements with the Atmospheric Lidar (ATLID) onboard EarthCARE which is a linearly polarized, high-spectral-resolution lidar (HSRL) system and represents a significant advancement in spaceborne lidar technology for studying cirrus clouds (Wehr et al., 2023). It allows us further to compare the cirrus cloud properties with the vertical updrafts and ambient aerosol loading.

*Code availability.*   Data description and example codes for handling the VFM data are available at https://asdc.larc.nasa.gov/ (NASA, 2025a). The MATLAB codes for drawing the plots in this paper are available from Zenodo at https://doi.org/10.5281/zenodo.15063832.

*Data availability.*   The CALIPSO data, including VFM used in this study, can be obtained via https://subset.larc.nasa.gov/calipso/login.php (NASA, 2025b, login required). The reanalysed data of cirrus parameters in different month in the years 2014 and 2018–2021 are available from Zenodo at https://doi.org/10.5281/zenodo.15063832.

*Author contributions.*   QL collected and analysed the data and wrote the manuscript with help from SG. Both authors discussed the results and findings and contributed to finalizing the manuscript.

*Competing interests.*   The authors declare that they have no conflict of interest.

*Acknowledgements.* This project has received funding from Horizon Europe programme under Grant Agreement No 101137680 via project CERTAINTY (Cloud-aERosol inTeractions & their impActs IN The earth sYstem) and the ESA-funded project ACtIon4Cooling (Aerosol Cloud Interaction for Cooling). Furthermore, this research has been supported by the DLR internal funding within the MABAK project (Innovative Methoden zur Analyse und Bewertung von Veränderungen der Atmosphäre und des Klimasystems). We thank the NASA Langley Research Center Atmospheric Science Data Center (ASDC) and CALIPSO science team for making the data available for research. This paper has greatly benefited from an internal review by Christof Beer (DLR, Germany).

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

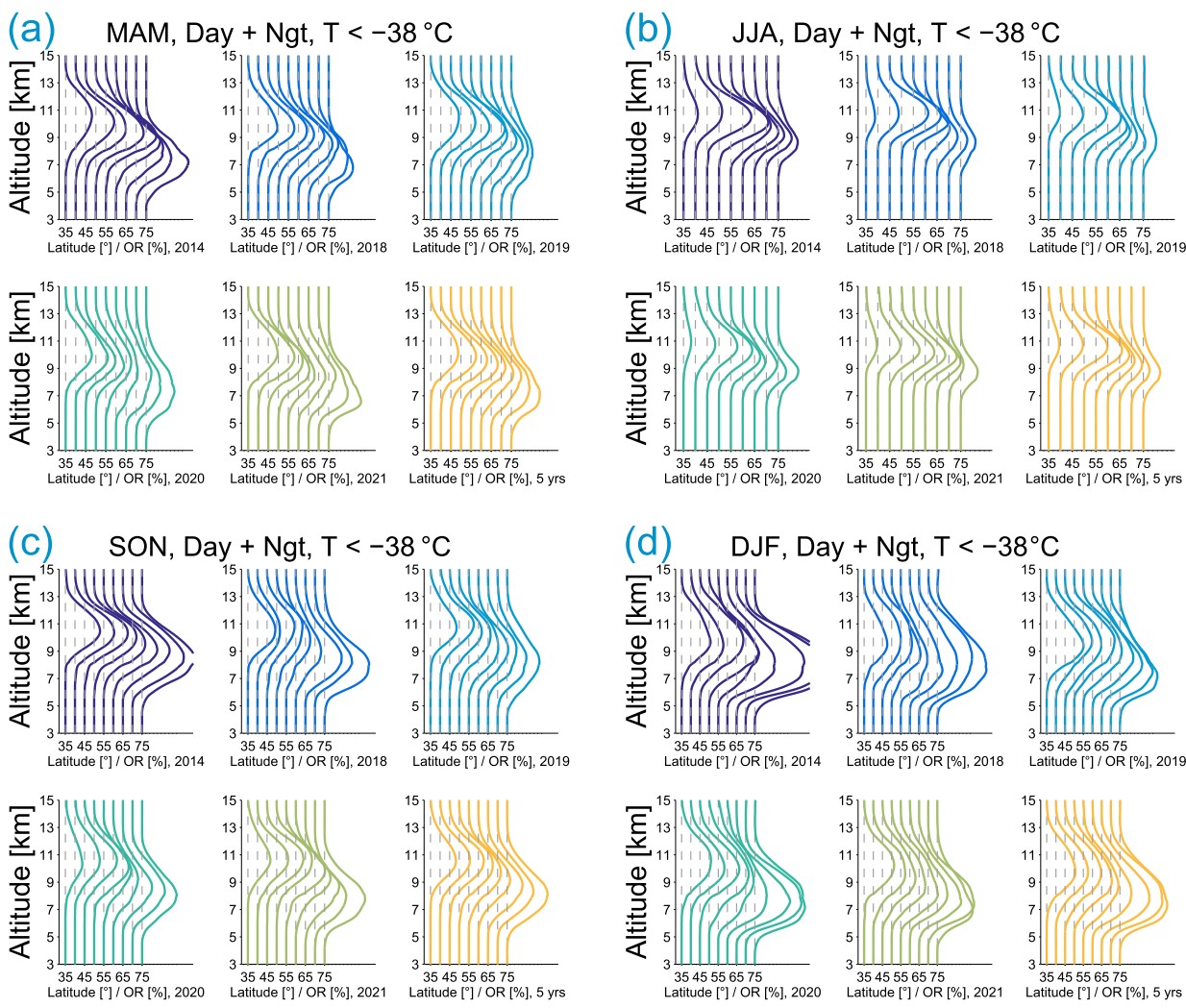

**Figure 1.** Profiles of occurrence rates (ORs) of cirrus clouds derived from the lidar measurements of CALIPSO in each 5-degree latitude bin from 35–80°N in different years (2014, 2018–2021) and the composite mean values from the 5-year observations. The vertical dashed lines indicating the zero levels of ORs and the corresponding profiles of cirrus ORs for each 5-degree latitude bin are offset by 3% for clarity. The ORs are shown in different seasons (MAM: Mar–May; JJA: Jun–Aug; SON: Sep–Nov; DJF: Dec–Feb). We note that the results of the DJF in 2014 and 2018 are derived from the observations only in Jan and Feb 2014 and 2018, and the rest DJF from Jan and Feb in the corresponding year and Dec from the previous year.

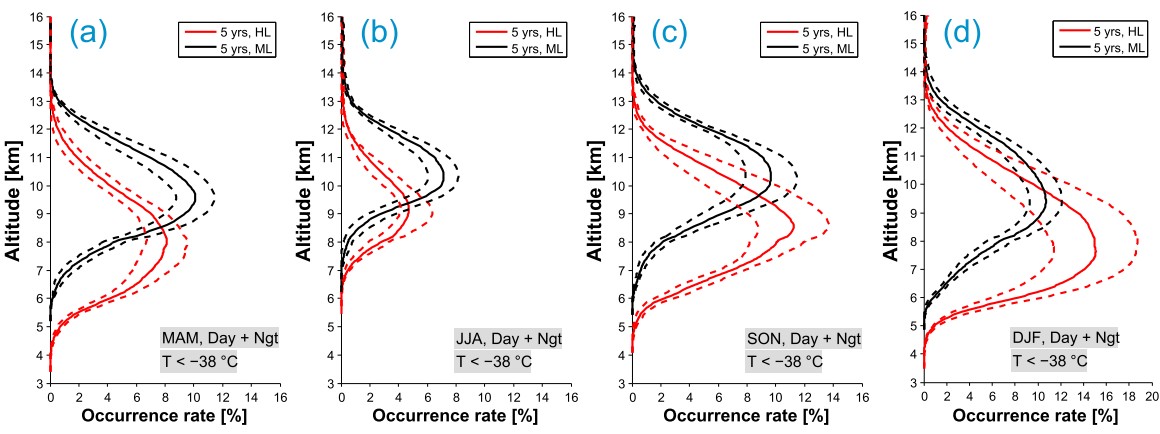

**Figure 2.** Profiles of occurrence rates (ORs) of cirrus clouds calculated from the composite 5-year lidar measurements of CALIPSO in different seasons at high-latitudes (60–80°N; 30°W–30°E) and midlatitudes (35–60°N; 30°W–30°E), respectively. The mean values of ORs are shown in solid lines and the standard deviations in dashed lines. The results at high- and midlatitudes are shown in red and black, respectively. Please note the different scales of x-axis for the winter results in the right-most panel.

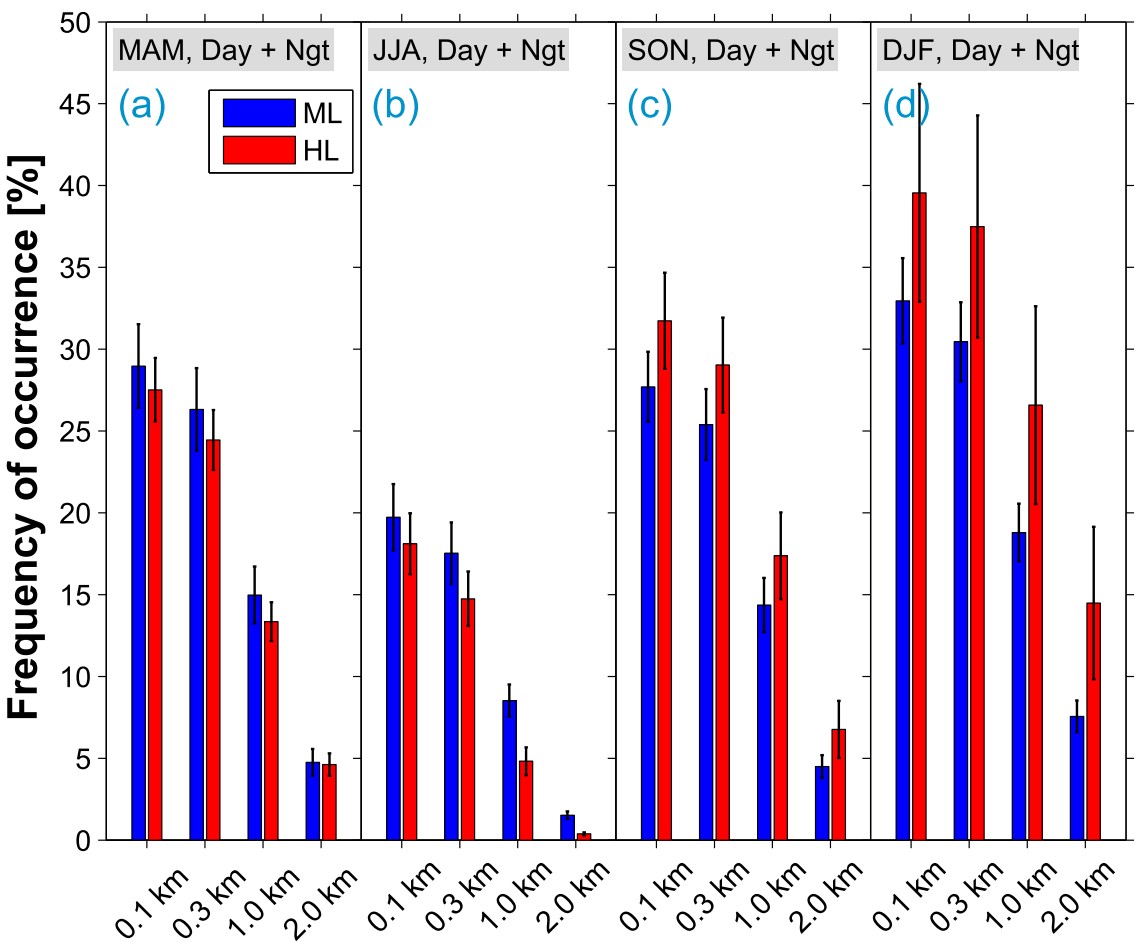

**Figure 3.** Frequency of occurrence of the geometrical thicknesses according to the definitions with cloud thicknesses larger than 0.1, 0.3, 1.0, and 2.0 km, respectively. The composite mean values are determined in different seasons (see the caption of Figure 2 for details) at high- and midlatitudes which cover the areas of latitudes from 35–60°N and from 60–80°N, respectively, with all the lidar measurements in years of 2014 and 2018–2021. The results in spring, summer, autumn, and winter are shown in the panels from left to right. The results derived from HL and ML are shown in red and blue, respectively, along with the corresponding errorbars.

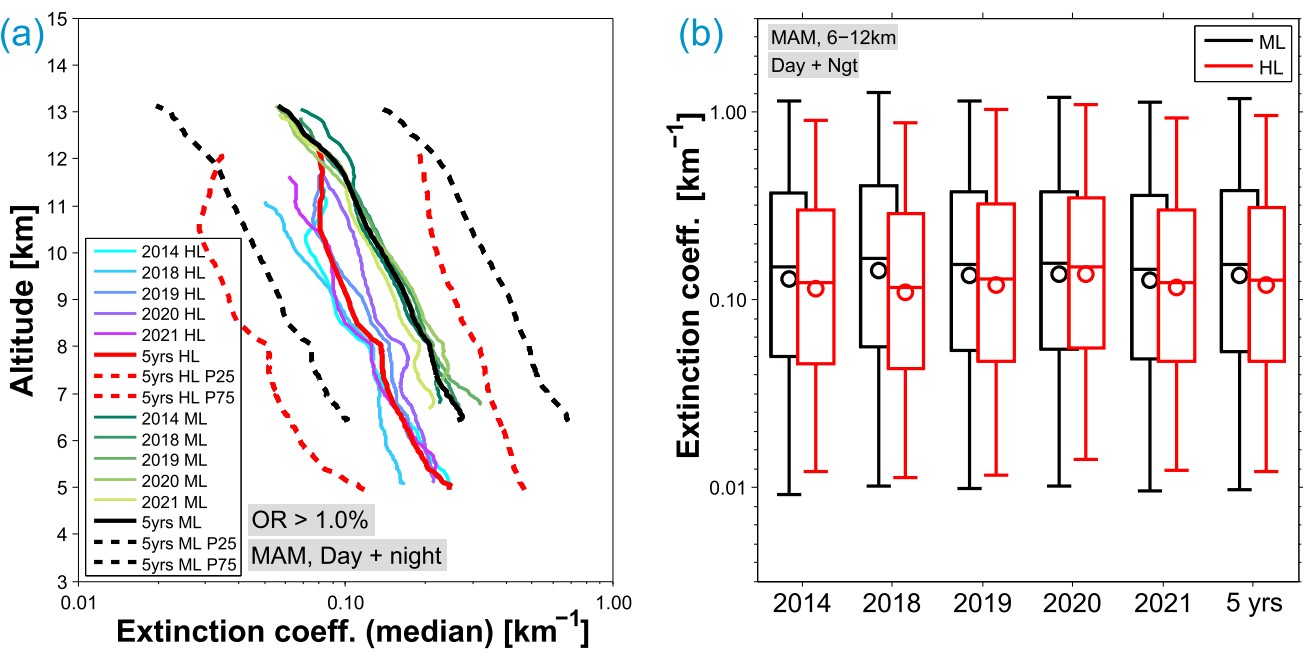

**Figure 4.** Left panel: Profiles of the extinction coefficient medians of ice crystals within cirrus clouds observed with CALIPSO in spring (March–May) in years of 2014 and 2018–2021 as well as all 5 years combined. The profiles of the 25th and 75th percentiles from the composite of 5-year data are also added (in dashed lines). Data with the cirrus occurrence rates less than 1.0% are ignored; Right panel: Box plot representations of the extinction coefficient distributions in different year and the composite values of all the 5 years. The results in the high-latitude regions are shown in red and in the midlatitude regions in black. Boxes represent the 25th–75th percentiles of the dataset (top and bottom). Solid lines through the corresponding boxes stand for the medians and circles for the means. Whiskers indicate the 5th and 95th percentiles.

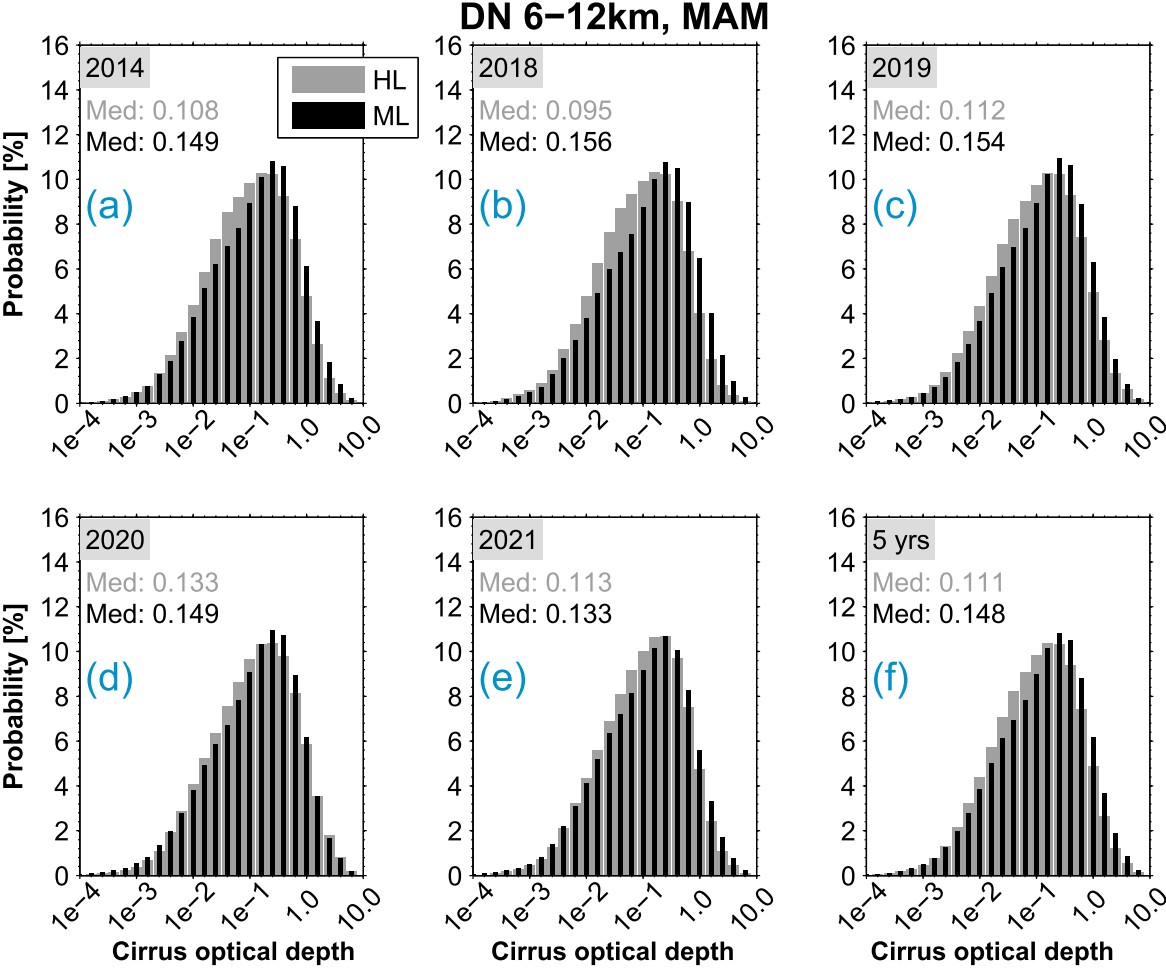

**Figure 5.** Histograms of cirrus optical depth observed in spring (March–May) in years 2014 and 2018–2021 in the high-latitude (gray) and midlatitude (black) regions, respectively. The corresponding medians for each case are indicated in the inset.

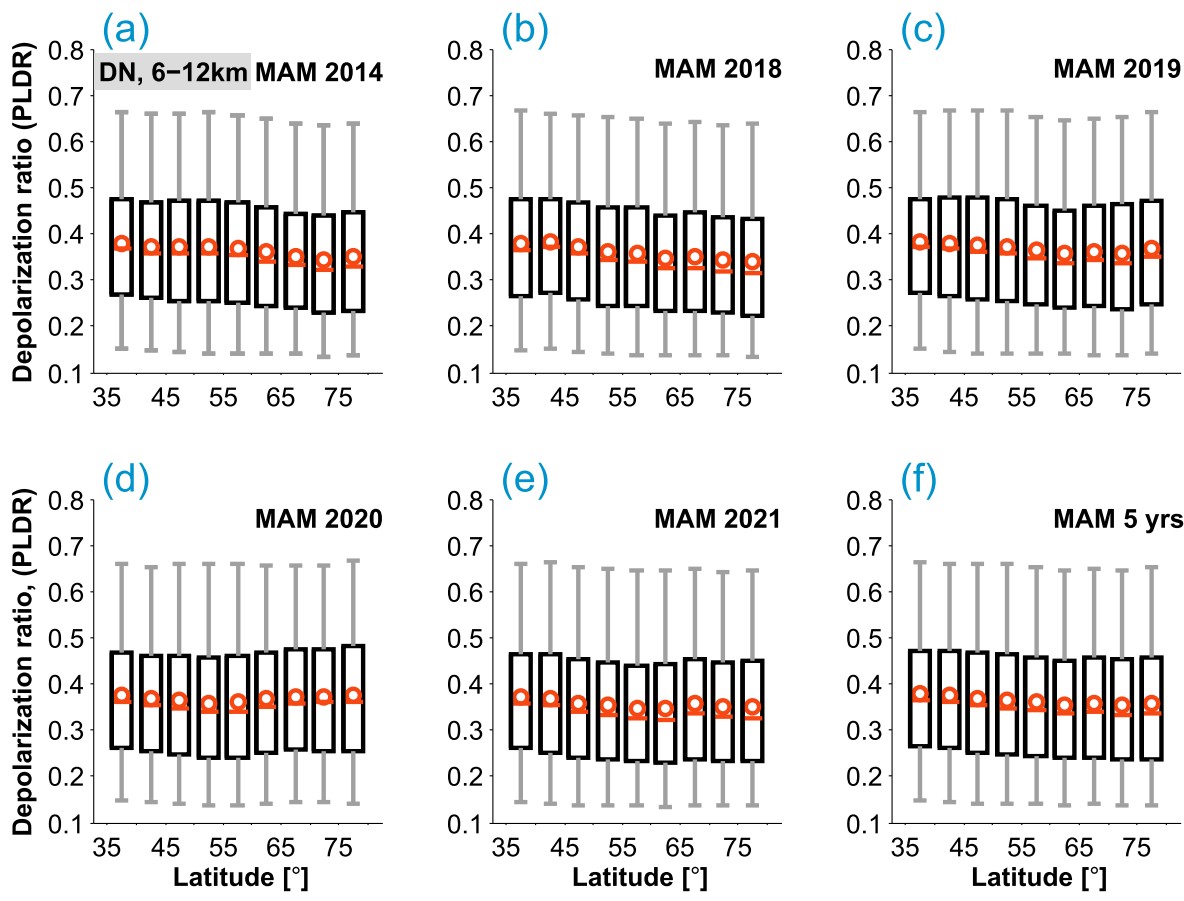

**Figure 6.** Box plot representations of particle linear depolarization ratios (PLDR) of cirrus clouds in each 5-degree latitude bin from 35–80°N in spring in the years 2014 and 2018–2021 as well as the composite results from all 5 years combined. The descriptions of the box plot can be found in the caption of Figure 4.

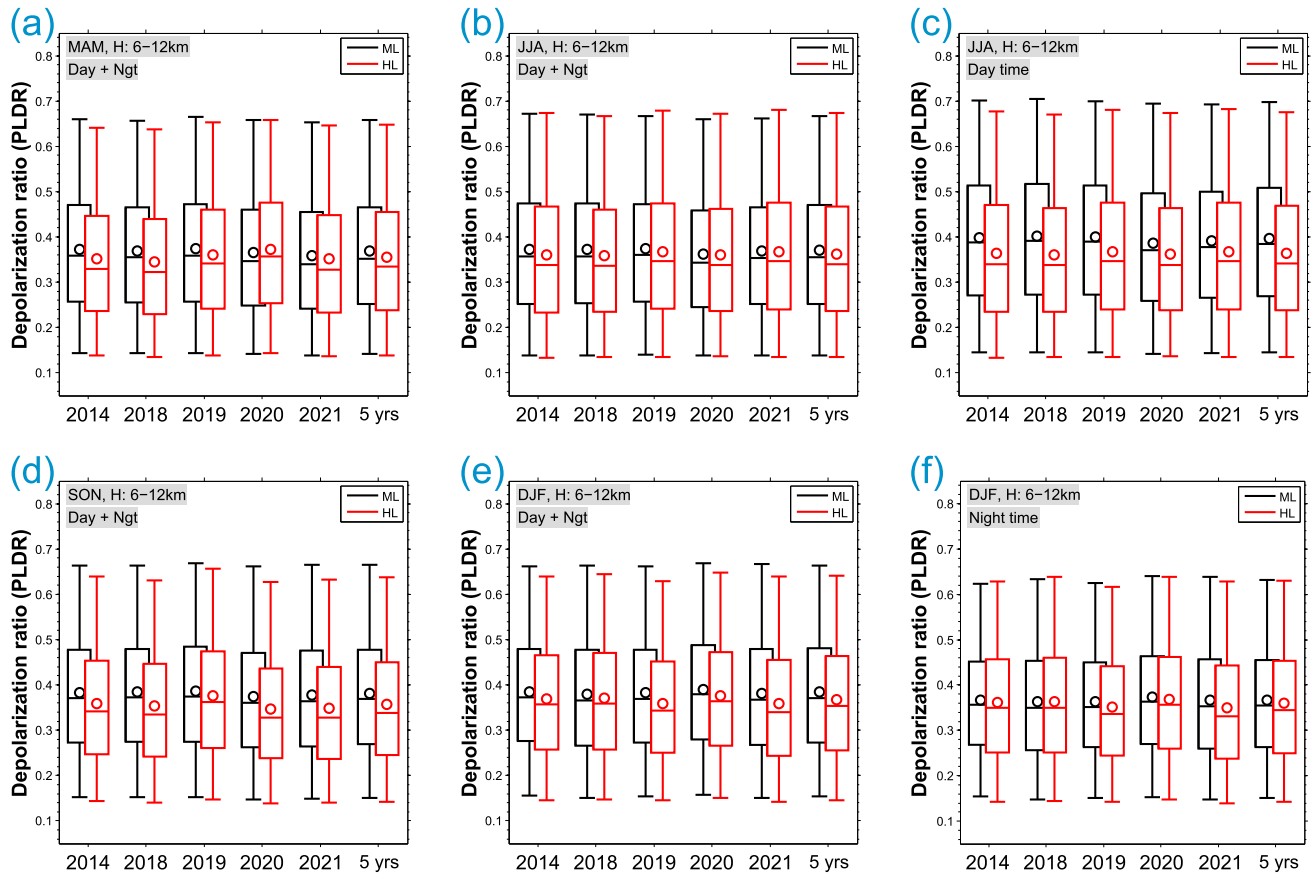

**Figure 7.** Box plot representations of particle linear depolarization ratios (PLDR) of cirrus clouds in different seasons in the years 2014 and 2018–2021 as well as the composite results from all 5 years combined (in panels a, b, e, and f). The results in the high- and midlatitude regions are shown in red and black, respectively. Boxes represent the 25th–75th percentiles (i.e. from bottom to top of the boxes). The descriptions of the box plot can be found in the caption of Figure 4. Considering the influence of polar day and night in summer and winter, respectively, on cirrus cloud properties, we also show the results from only the daytime measurements in the summer months and from only the night-time measurements in winter for a fair comparison in different latitudes (in panels c and f).

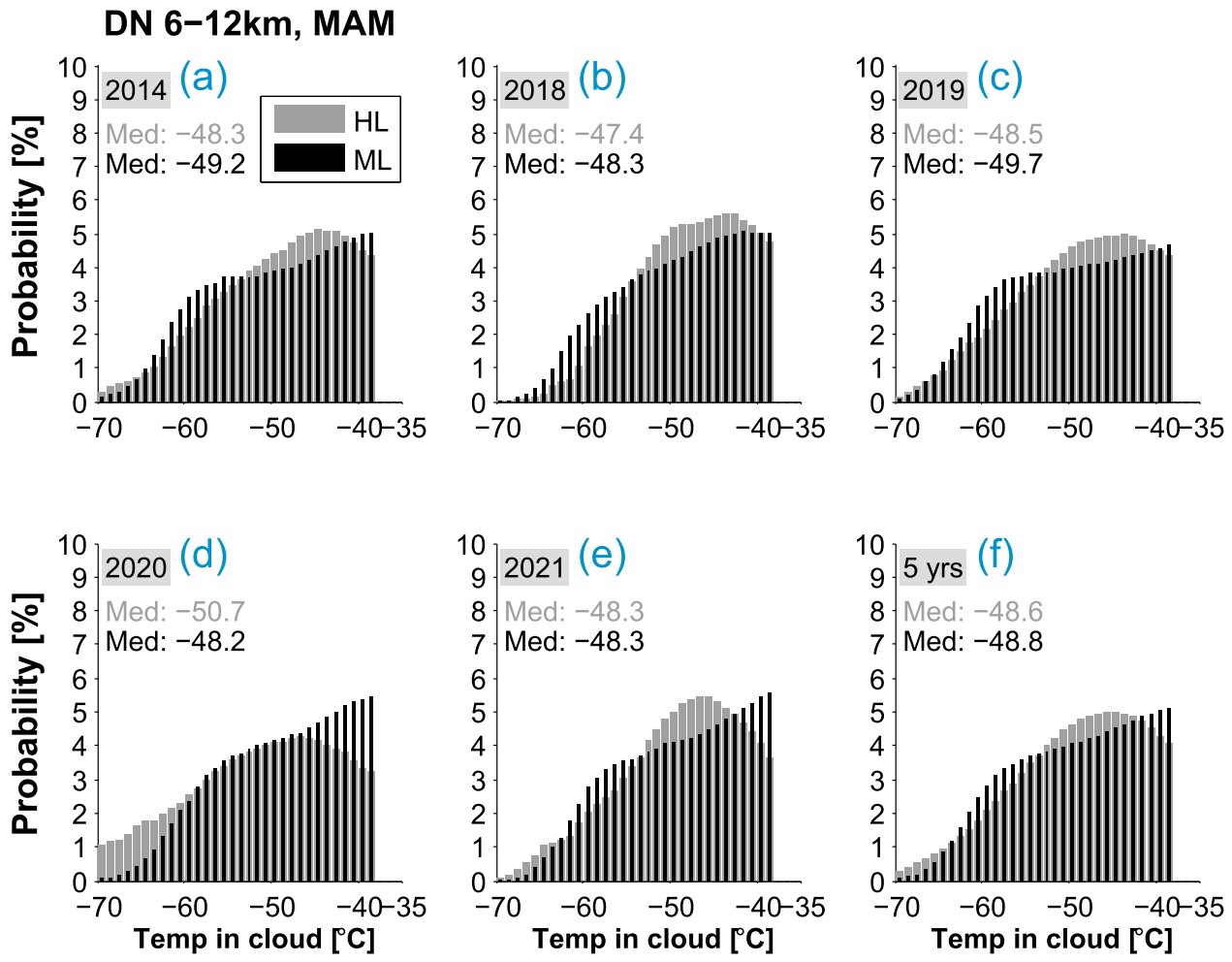

**Figure 8.** Comparison of temperatures inside cirrus clouds within different latitudes in spring. The histograms at HL are shown in gray and at ML in black with the medians indicated in the inset. In general, their distributions show a consistence in different years at different latitude domains, respectively. The formation of cirrus clouds at ML took place with the maximum occurrence at temperatures of -38°C, with a "shoulder" for temperatures down to -60°C, and a long tail extending to lower temperatures. Cirrus clouds at HL, however, occurred with the maximum probability at ∼-45°C and also with a long tail extending to lower temperatures.