# Peer review of "Lidar observations of cirrus cloud properties with CALIPSO from midlatitudes towards high-latitudes"

_EGUsphere, 2025_

## Author Comment (AC1)

RC2: 'Comment on egusphere-2025-2052', Eleni Marinou, 26 Aug 2025 reply

Thank you to Eleni for her careful review and for the comments and suggestions on this manuscript. In the following, the Referee's questions and comments are repeated in black and our responses follow in blue.

This article by Li and Gross presents an analysis on statistics of cirrus cloud properties at Mid and high latitudes, and discusses the different findings per season, altitude, temperature, and aerosol abundance. The results are new, and the study is relevant to the objectives of the journal. The work is complete, scientifically accurate, and significant, and the manuscript is well-written and well-structured. Overall, the study is suitable for publication. Certain sections could benefit from some additional clarifications, as described herein.

**General comments:**

Smoke layers from Canadian fires were frequently detected from CALIPSO during the years of the study. It would be interesting to include a discussion about the possible effect of the elevated smoke layers in the stratosphere on the ice in these altitudes in the 2 domains.

→ Thank you for the general comments. Indeed, smoke layers from Canadian wildfires were frequently detected with CALIPSO and with ground-based lidars in Europe. It has become a hot topic in the relevant research of aerosol-cloud interaction since the larger amount of smoke aerosols lifted into the UTLS regions can act as INPs to trigger cirrus formation and modify cloud optical and microphysical properties. In general, smoke provides surfaces for heterogeneous ice nucleation, suppressing homogeneous nucleation, which leads to the formation of fewer, larger and more irregular (non-spherical) ice crystals. In addition, smoke aerosols acting as INPs can trigger ice nucleation at higher temperatures and lower supersaturation leading column-like and bullet-rosette crystal formation. Consequently, this increases PLDR and tends to reduce extinction efficiency (and optical depth). However, due to different backgrounds in the 2 domains of high- and midlatitudes, the influence (or the contribution of the influence) of biomass burning smoke on cirrus clouds can be very different. Smoke transported into the very cold and stable UTLS regions at high latitudes (and Arctic) can suspend very long, providing a persistent INP reservoir for cirrus formation. Smoke at midlatitudes is normally episodic and mixed often with other aerosols and airmasses. The competition with other local aerosols, like dust, marine aerosols, and anthropogenic pollution aerosols, can dilute the smoke contribution to ACI.

Consider including in the abstract the information that the cloud statistics of this work focus on temperatures <-38C, excluding ice observations above these temperatures.

→ The information is added.

In section 4.3, it is not clear what the contribution of this study is to this discussion. I suggest revising the text to make your results clearer in relation to or in addition to the past studies in this summary. The way it is written now could be part of the introduction of this paper.

→ Section 4.3 is a very important part to interpret the observed difference of cirrus cloud properties in the high- and midlatitude regions, although no simultaneous measurement of embedded aerosols is available for the current study. For our argument, the statistical distributions of aerosol particles, theoretically and observationally, from literatures are sufficient for the scope of the current study.

**Specific comments:**

Page 1, line 9: "The distributions of PLDR in each 5-degree latitude bin show a general decrease with increasing latitude": Suggestion to add the physical meaning of increased PLDR.

→ The physical meaning of increased PLDR is added.

Page 4, line 124: "level 2 5-km cloud profile products": It is useful to include the version of the product.

→ "The Version 4" is included.

Page 4, line 125: "all the atmospheric entities": With this phrase, one may be confused whether the product also includes information on the aerosols. Consider revising.

→ In the next sentences, we clarified that we distinguish cirrus clouds from other features including aerosol by using VFM as well as a temperature threshold (<-38°C).

Page 5, line 128: "...VFM...": It is useful to include the version of the product.

→ "The Version 4" is included.

Page 5, line 138: "mid-latitudes (35–60°N; 30°W–30°E) and high-latitudes (60–80°N; 30°W–30°E)": suggestion to include a figure with the map and the 2 domains (maybe in the appendix).

→ A map of the research area is added in the supplementary material.

Page 5, line 142: "in 5 years of 2014 and 2018–2021 are analyzed": Please include a short explanation why you exclude the other good CALIPSO years (2007-2017).

→ It is clarified that "the choosing of 2014 is due to the potential cross comparison study between satellite and airborne measurements during the ML-CIRRUS field campaign ...". The years of 2018-2021 are chosen for covering 2 non-COVID years, a heavily COVID-influenced year of 2020, and 2021, a year with relatively mild COVID-19 impacts.

Page 5, lines 156-158, and Table 1: I suggest considering excluding the 2nd digit after the decimal point (statistically not significant).

→ They are revised accordingly.

Page 6, line 6: "variations in the altitudes with the maximum ORs along the latitudes are discernable, showing the largest values in summer": suggestion to change as "showing the largest altitude values in summer".

→ They are revised accordingly.

Page 6, line 192: "which is related to the larger variabilities in humidity at HL than at ML and is consistent with a recent model study showing larger INP effects on cirrus at higher latitude": Isn't the largest variability of temperature also a significant contribution for HL ORs also?

Temperatures play a crucial role in cirrus formation. Is the temperature variability larger at HL? I did not find any reference to back up that.

Page 7, line 202: "show that the thickest cirrus clouds formed in winter and the thinnest ones in summer": I am not sure I see this in the plot, as overall Winter has the highest CR for thin clouds also. Can you rephrase this part to make it clearer?

→ From Figure 3, we can clearly see that the probability of occurrence with the geometrical depth larger than 0.1 km (or 0.3, 1.0, and 2.0 km) is the largest in winter and smallest in summer.

Page 7, line 210: "presumed to be smaller at HL than ML": Is this correct? It seems higher at HL.

→ Thank you to point it out. Aviation density is much lower at HL, which means the contribution of aviation impact on cirrus at HL is lower. So the influence due to aviation reduction during COVID is presumed to be smaller at HL.

Page 7-8, section 3.2. Consider including a sentence on why you concentrate in Spring among all the seasons, if there is a reason of interest to the reader.

→ It's mentioned that the datasets we analyzed in this study cover pre-COVID years, a strongly COVID-influenced year, and a moderate COVID-influenced year. The differences in cirrus properties under COVID-impact at HL and ML are the most pronounced in Spring. Nevertheless, the distribution of extinction in other seasons are also shown in the supplementary material. The descriptive texts are added.

Page 8, line 234: "indicating smaller and fewer ice crystals at higher altitudes": suggestion to rephrase to and/or, and it could be one or the other also.

**→ Thank you. Revised.**

Page 8, line 237: "This is closely linked to the dominant formation processes of ice crystals depending on temperature and relative humidity over ice (RHi). "This is also closely linked with the most frequent abundance of INP in these altitudes at ML. Consider revising this part to avoid confusion.

→ The discussions are revised.

Page 13, line 400: "more irregular": more in comparison to what? Please enhance this sentence for clarity.

→ The sentence is revised.

Page 15, line 452: "we compare aerosol concentrations at different latitudes": Where is this comparison shown? Consider adding a plot of the aerosol concentrations or revising the sentence by e.g., "we compare ice crystal concentrations in regions of different aerosol concentrations as reported in previous studies".

→ The comparison of aerosol concentrations at different latitudes is widely recognized in terms of statistics and can be found easily in literature. The sentence is revised accordingly.

Figure 1: If possible, add a scale indicating the magnitude of the OR[%] in these plots.

→ The span between the two adjacent dashed lines indicates 3% of OR. The information can be seen in the caption of Figure 1.

Figure 3: Based on these occurrences, ML has more clouds than HL in spring and summer. This is surprising. Can you include a comment in the manuscript on this, maybe backing up the findings with past studies on cloud abundance, or is this a new finding? Also, it would be useful to include a description or equation on how these occurrences are calculated.

- → Cirrus cloud occurrence is generally higher at midlatitudes on an annual mean basis. But, it can become comparable to or higher at high latitudes compared to midlatitudes in autumn and winter due to enhanced moisture transport from lower latitudes and convective activity. From the CALIPSO measurements of cirrus clouds, there are more clouds at ML than HL in spring and summer (e.g. Sassen et al., 2008; Nazaryan et al., 2008; Stubenrauch et al., 2013; Gasparini et al., 2018). So, it is not a new finding but consistent with previous studies.
- → We calculate the frequency of occurrence with cloud thickness larger than a threshold following *Li* and *Groß* (2021). The full description of the calculation is added.

**Typos:**

Page 5, line 138: midlatitudes: mid-latitudes

Page 6, line 160: discernable: discernible

Page 11, line 342: exam: examin

Figure 1: 2014 (2018): 2014 and 2018

→ Thank you. They are revised.

Figure 3: occurrence frequency: occurrence rates?

→ The term 'occurrence rate' in this manuscript is defined as the ratio of number of lidar profiles where cirrus is detected at altitude z to total number of valid lidar profiles at that altitude. The term "occurrence frequency" is used to describe the probability of the occurrence according to the definition with cloud thickness larger than some threshold. To avoid misunderstanding, it is changed to "frequency of occurrence".

Citation: https://doi.org/10.5194/egusphere-2025-2052-RC2

---

## Author Comment (AC2)

**Response to Review RC1 by Referee #1.**

Thank you to the anonymous Referee #1 for his/her careful review and for the comments and suggestions on this manuscript. In the following, the Referee's questions and comments are repeated in black and our responses follow in blue.

The authors present spaceborne (CALIOP) lidar observations of cirrus clouds and compare the measurements performed at midlatitudes and high latitudes. This is a good contribution to cirrus research at mid to high northern latitudes. However, several parameters are not well defined. The discussion of the results needs to be improved. Uncertainty ranges need to be given.

Minor revisions are needed.

P1, line 10: We need a clear definition of the 'effective optical depth'. Why do you introduce 'effective'?

→ Thank you for pointing out the parameter. Here we calculated the true or geometrical optical depth from lidar. In the revised manuscript, 'Optical depth' will be used accordingly, as used in literatures.

P2, line 57: Here, you could add the recent MOSAiC publication on smoke and cirrus (Ansmann et al., ACP; 2025).

→ Yes, both companion MOSAiC papers are added.

P3-P4: Most of the information on page 3 and page 4 are not needed and could be left out. The shorter the introduction the better. Come to your point of research in 1.5-2 pages!

→ The statements from previous studies are the main motivation of the current study. Anyway, the introduction is reduced accordingly.

P4, Eq.(1): Why do you use 'effective'? It is simply the cirrus optical depth! In this context you may already explain how you get the single-scattering extinction coefficient sigma-ci. How did you correct for multiple scattering? Maybe you use the backscatter coefficient, obtained from the CALIOP observation with a lidar ratio that considers multiple scattering? What lidar ratios does the CALIOP team assume in their retrieval of the cirrus backscatter coefficient? And what lidar ratio do they assume in the multiplication of the backscatter coefficient to obtain the single scattering extinction coefficient? This is important information that is missing in this lidar paper on cirrus optical properties.

More general, how is the extinction coefficient profile determined? How large are the uncertainties when using the Klett technique? I speculate, probably much larger than 50%!

→ Sorry for the misleading. Yes, it should be simply cirrus optical depth. The CALIOP lidar onboard CALIPSO is a backscatter lidar and actually measures attenuated backscatter which contains information on cloud backscatter and extinction. The CALIPSO team uses a constrained Klett-Fernald inversion of the lidar equation. With an assumption of a constant lidar ratio for different entities, which depends on types and subtypes as well as on the centroid temperatures in the layers (e.g. for 35 to 21 sr for cirrus clouds in Version 4),

backscatter coefficient can be determined by solving the lidar equation and extinction coefficient can then be converted from backscatter coefficient (with assumed lidar ratio). CALIPSO delivers extinction as if it were single-scattering with correction from multiple-scattered lidar signal. The assumption of lidar ratio is combined with a multiple-scattering correction factor (e.g. 0.6 for cirrus) to account for the effect of photons scattering more than once. Unlike state-of-the-art EarthCARE, CALIPSO cannot independently measure both extinction and backscatter to calculate the lidar ratio of cirrus which can vary significantly depending on ice crystal size, habit, and ice water content. The assumption of lidar ratio will introduce uncertainties into the retrieved extinction, which dominates the total uncertainty. Furthermore, the correction of multiple scattering, signal noise (especially for thin cirrus), and calibration also introduce uncertainties. In total, the relative uncertainty for cirrus can be 25%, which is improved in Version 4 compared with the previous versions.

P4, line 123: CALIOP is nadir pointing? It measures at an off-nadir angle of 3°! So, it is off-nadir pointing! One could briefly explain why an off-nadir pointing is selected.

→ The referee is right that CALIOP was operated at a 3° off-nadir angle from late 2007 onwards. The aim to change the lidar to off-nadir pointing is to reduce the influence of specular reflection from horizontally oriented falling ice crystals, to prevent errors in depolarization measurements, and to improve the accuracy of cloud property retrievals.

P5, line 149: 'Occurrence rate'? You probably mean: Frequency of occurrence! The word 'rate' points to occurrence per second. More common is to use 'Frequency of occurrence', i.e., number of cirrus layers occurring within a given latitudinal belt within a given season.

→ Here, the term 'occurrence rate' is defined as the ratio of number of lidar profiles where cirrus is detected at altitude z to total number of valid lidar profiles at that altitude. Hence, it is altitude-resolved with vertical distribution. That is what we want to express. While, 'Frequency of occurrence' may consider the entire profile for overall presence or others.

P5, line 158: You write: The altitude ranges in which cirrus formed... can be seen in Figure S1. How do you know where cirrus formed? I speculate that you often detect just virga segments far below the height where ice crystal nucleation (in situ cirrus formation) took place. Please, be more clear in this respect. You may also use 'height interval' as an alternative to 'height range'. In conclusion, you mean the height interval in which cirrus segments were found... or cirrus clouds occurred. Please state that clearly!

→ We calculate cirrus occurrence rate along altitude within different seasons for each 5-degree latitude bin. In order to clearly illuminate the main information of the results, we plot the altitude ranges in which cirrus occurred (only with OR > 1%) in Figure S1. We use 'altitude range' to refer to the overall vertical span with cirrus formation, like 9-12 km. Of course, CALIPSO measurements have height resolutions, but still it is the right term that should be used. 'Height interval', however, emphasizes the steps or spacing within the range.

P6, lines 166-168: The extent, size, or depth of the height interval in which you detected cirrus layers mainly depends on the size or vertical extent of the virga zones, i.e., by the base heights of the detected virga layers. That means: 'Height range of cirrus formation' is definitely misleading wording. This hold for the entire page 6.

→ Due to the detection limit, CALIPSO may not see virga when they are weak. Small virga streaks can be smoothed out when the signal profiles are horizontally averaged over 5-km at Level-2. In addition, virga may be flagged as aerosol if CALIPSO can see them. Naturally, we can't deny the fact that the detection of virga will lead to an extended cloud base. The CBH detected by CALIPSO has been widely compared with ground-based ceilometers in literatures (e.g., Lu et al., 2021), indicating that CALIPSO with the help of VFM can retrieve CBH efficiently. Additionally, we applied here a temperature mask (< -38°C), which can likely reduce the impact of virga since they normally appear with higher temperatures. Of course, we indicated in the manuscript that we consider only the scenarios with OR > 1.0%, which shrinks the "altitude range of cirrus formation". With the caveat, the term is not misleading.

P7, I 197: The geometrical thickness of a cirrus cloud is obtained from the knowledge of the base and top height of a given cirrus layer. Your definition is confusing (line 197): the geometrical thickness of cirrus cloud is defined as the vertical distribution of cirrus clouds at different latitudes. What do you mean here? What do you want to tell us?

→ Sorry for the confusion. The geometrical thickness of cirrus clouds is calculated as the sum of altitude extension (vertically) with cirrus formation despite of the multiple layers of cirrus clouds. That is how cirrus thickness defined. With different thresholds in the definition (larger than 0.1, 0.3, 1.0, and 2.0 km), the occurrence frequencies can be calculated, respectively, which are shown in Figure 3.

Figure 3 is misleading. You need to improve the figure, you have to write clearly: >0.1km, >0.3km, >1.0 km, >2.0km. Why do you not show histograms? If my interpretation of Figure 3 is correct, most detected cirrus layers have thicknesses from 0.3 to 2 km. I further speculate that thin cirrus layers with 100 to 200 m thickness are probably sublimating virga structures at all. Cirrus nucleation cells show immediately cirrus thicknesses of >300m within 20-30 minutes after nucleation of first ice crystals, resulting from ice growth and sedimentation processes.

→ I don't know why Figure 3 is misleading. The information for the definitions with different thresholds is clearly given in caption. Of course, your interpretation of Figure 3 is not correct. And I don't agree with you on your speculation that thin cirrus layers (thinner than 200 m) should be virga. And we categorize the geometrical thickness of cirrus into four groups with the thickness larger than 0.1 km or other numbers, not equal to 0.1 km. The conditions with cirrus thickness less than 0.1 km will be considered to be cloud-free.

P7, line 214-215: better write 39%, 13.5% and 15.5%.

→ Thank you. The numbers are revised.

P7, line 219: 'rate of decrease' is misleading! You simply mean 'decrease'

→ Really? As I understood that the cirrus extinction quantifies how much the intensity of light decreases per unit distance as it passes through a cirrus cloud layer. So it is not the total decrease, but the rate of decrease with distance light traveled in the medium.

P8, line 223-231: In such a cirrus paper, we need clear information on the CALIOP cirrus data analysis! As I already asked above: how is the extinction coefficient obtained from the CALIOP raw data. What lidar ratios did they use to obtain the Klett solutions for the cirrus backscatter coefficient? Please, provide numbers here. The lidar ratios, used in the Klett procedure, consider multiple scattering. Afterwards, the multiple-scattering-corrected cirrus backscatter coefficients is obtained. What lidar ratio did they use next to obtain the respective single-scattering extinction coefficient? This extinction profile can then finally be used to calculate the cirrus optical depth. It is simply not sufficient here to provide the reference Vaughan et al. (2009). At the end, we need to know how large the uncertainty in the used cirrus optical depths are! The use of 'effective optical depth' is confusing for all non-lidar scientists.

→ Thank you. The term 'effective optical depth' will be revised. The processes of extinction retrieval are described and the estimated uncertainty is also given from literatures.

P8, line 232: The discussion of extinction coefficients shown in Figure 4 must include the uncertainty in the CALIOP retrieval products. The solutions for the extinction coefficients could be compared with the ones shown for the MOSAiC cirrus clouds (Ansmann et al., 2025) and also with other studies mentioned in that paper.

→ The uncertainty in extinction retrievals has been discussed above. The uncertainty contributions from different sources are summarized in Table 2 from literatures. The relative uncertainty from the CALIPSO data of Version 4 is about 25% (Young et al., 2018). To provide more information, we also add the 25th and 75th percentiles (from all 5 years combined) along with the medians in Figure 4(a). The distribution of derived extinction (including medians as well as the 5th, 25th, 75th and 95th percentiles) can be seen in Figure 4(b).

P9. Yes, there may be a daytime vs nighttime difference in the cirrus extinction coefficient. But daytime CALIOP data are much noisier than nighttime data... and the Klett solutions may be much more uncertain for daytime cirrus cases than for nighttime cirrus cases. Therefore, we need the CALIOP extinction uncertainty information! Trustworthy uncertainty numbers start at 50%! Uncertainties are larger for noisy daytime data than for less noisy nighttime data.

→ The referee is absolutely correct on this point. The SNR were lower during the daytime than night time which may contribute roughly 10-15% uncertainty to extinction retrievals, mentioned in the new table.

P10: In Fig.6 (and S6-S8), I do not see a clear change of the particle depolarization ratio with latitude, only a weak tendency. The variability indicated by the size of the boxes and bars is so large that a clear dependence is not obtained.

A significance test is added. Based on the Mann-Kendall (MK) significance test, there are clear decreasing trends in the medians as well as 25th percentiles and 75th percentiles of PLDR along latitudes from 35 to 80 degree with the p-values much smaller than 0.001. The exercises have been done to the values in different seasons from different years as well as from all 5 years combined. That means there is a clear change of PLDR with latitude. Only exception is found in the case in Spring (Mar-Apr-May) 2020 when the civil aviation was significantly reduced in Europe (midlatitude) due to the COVID-19 pandemic as well as due to the unusual strong stratospheric polar vortex. It's partially due to the reduction of aviation impact on cirrus PLDR, which has been reported in our publication (Li and Groß, 2021). See the following figures for the results of significance test for the combined 5 years of data in different seasons.

Figure R1. The dependence of PLDR on latitudes ( $25^{th}$ ,  $50^{th}$  = median,  $75^{th}$  percentiles of the datasets representing the entire datasets). The results in different seasons are shown in the different panels. The

slopes of the tendency as well as the p-values for the significance test are calculated and indicated in the inset.

Also the differences (HL vs ML depolarization ratios) are quite small, and if day vs nighttime observations show strong differences one must be careful in the interpretation of the results because of the different background noise impact on nighttime vs daytime products.

→ From the previous answer, the MK test shows there are decreasing trend in PLDR with increasing latitude. Consequently, the values of PLDR at ML and HL are statistically different at a high significance level. Yes, the day- and night-time observations show differences in the distributions. That is why we also compared only daytime observations in summer and only night-time observations in winter, considering the influence of polar day and polar night. Nevertheless, the comparisons show nearly the same tendency at ML and HL under the same conditions of solar elevation.

Please, provide your hypothesis on the link between aviation (stronger at daytime), crystal size, and depolarization earlier, i.e., already on page 10. What about the impact of shape.... plates vs columns ... on the depolarization ratio? Could be discussed as well.

→ It's mentioned in the Introduction that PLDR depends on the shapes of ice crystals from the previous studies. To explore the dependence on PLDR on ice crystal properties in more details, the shape ratio (or aspect ratio) (=length/diameter of crystal) was introduced. In general, the PLDR values increase with the increasing shape ratio under the same conditions, i.e., column-like crystals tend to produce higher PLDR than plates. However, other factors may also play a crucial role on PLDR, like, particle size, orientation, surface roughness, and the background conditions (including temperature and humidity). Therefore, there is no one-to-one direct correlation between PLDR and geometrical shapes of ice crystals. From the current measurements, we have no information on the ice crystal shape and cannot proceed further. The topic is, of course, valuable to explore in more details in the future.

P13: What about the impact of wildfire smoke on cirrus evolution? Should be included in the discussion. Furthermore, long-range transport of aerosol in the (upper) free troposphere occurs everywhere, at all latitudes. So, why should there be a decrease of the aerosol content in the upper troposphere with increasing latitude?

→ Wildfire smoke has been shown to have a strong impact on cirrus evolution. As one of the aerosol sources they bring soot, organics and mineral particles into air that can act as INPs for cirrus formation. The process can increase heterogeneous nucleation and suppress homogeneous nucleation depending on the particle composition and concentration, which leads to changes in cirrus optical and microphysical properties. In the current manuscript, the general information of aerosol distribution is enough since we compare the correlations of cirrus properties with background aerosol loading in the sense of statistics.

P14: Conclusions should be compact and short. Again you write: Cloud formation shows a clear decrease with latitudes. Maybe!, but you observed cirrus features from the top to the virga bottom. The height distribution of cirrus formation cannot be derived from CALIOP cirrus observations.

- → Conclusions are revised accordingly.
- → I don't agree with the point that the measurements of cirrus features are from the top to the virga bottom. It is described in the manuscript that the data at temperatures above -38°C and with PLDR smaller than 0.1 or larger than 0.8 have been removed (see the plots below).

Figure R2. The cloud subtypes derived from the VFM products (Left panel); The derived PLDR of cirrus clouds with the distribution of them in the inset.

P14, lines 444-446: Again these temperatures are related to all cirrus observations, but not exclusively to locations of cirrus formation.

→ It is not clear if it is a question or comment.

P15, I 452: You do not have measured aerosol concentrations! You did not report any aerosol observation (including typing) for the upper troposphere. So this last paragraph presents mainly your speculative ideas, rather than clear observations. It seems to be that wildfire smoke plays a major role in the upper troposphere over the mid latitudes as well as over the high latitudes, especially since the strong Canadian wildfires in the autumn of 2017. However, all these statements are just hypotheses. Please state that clearly!

- → In the manuscript, we clearly point out that we compare cirrus cloud properties in different latitude domains with different aerosol concentrations reported in previous studies. The statistical distributions of aerosol particles, both theoretically and observationally, from literatures provide sufficient support for our conclusions for the scope of the current study. Unfortunately, direct measurements of aerosol loading in the background during the period of this study is not available for now, which, however, should be further explored in the future.
- → The referee is absolutely right that large amount of smoke aerosols lifted into the UTLS regions during wildfire outbreak. Smoke particles can act as INPs for cirrus formation at lower supersaturation, which enhances heterogeneous nucleation and suppresses homogeneous nucleation for competing available water vapor, and hence leads to changes in cirrus cloud properties. Nevertheless, the influence of smoke on cirrus clouds at different latitudes may vary significantly due to the difference background in the two domains.

Figure 1: dashed lines are shifted by 3%? What does that mean? a), b), c), d) is missing

→ Sorry for the confusion. It is simple that the reference lines (= zero OR) for each 5-degree latitude bin and the corresponding profiles of cirrus ORs are shifted by 3% (i.e., +3%) for better visualization. The caption is rephrased. The marks for different panels are added.

Figure 2: You mean 'Frequency of Occurrence'? a), b), c), d) is missing

→ No, it is "Occurrence rate". The labels for different panels are added.

Figure 3: Occurrence frequency! No longer OR. a), b), c), d) is missing

→ Yes. They are different parameters. To avoid misunderstanding, the term "frequency of occurrence" is used as in literature like *Ansmann et al.*, (2025).

Figure 4: a) and b) is missing. Uncertainty bars (SD bars) are missing. They would show how trustworthy the observed differences are.

→ It's mentioned above that Figure 4(a) intends to give a general comparison for the altitude dependence of extinction. Without error bars, it is clearer and more straightforward. Figure 4(b) gives the information in details on the distributions of cirrus extinction. As mentioned above that the estimated relative uncertainty in extinction is around 25% (*Young et al.*, 2018) in the CALIPSO Version 4 (*Young et al.*, 2018). To provide more information, we also add the 25th and 75th percentiles (from all 5 years combined) along with the medians in Panel (a). The labels for different panels are added.

Figure 5: a), b), c), d), e), and f) is missing. The values could be compared with other studies (e.g., Ansmann et al., 2025).

→ The labels for different panels are added. The values of derived cloud optical depth of the current study are comparable with previous studies (like, *Ansmann et al.*, 2025).

Figure 6: a), b), c), d), e), and f) is missing. Very harmonic data, a tendency is visible...

→ Thank you. The labels for different panels are added.

Figure 8: a), b), c), d), e), and f) is missing. What can we learn from a distribution of temperatures linked to the detection of cirrus filaments and structures. The shown temperatures do not show cirrus formation temperatures.

→ The labels for different panels are added. It's reported that temperatures determine ice crystal formation pathways and further affect crystal habit and shape. Furthermore, the optical properties of cirrus cloud (e.g., PLDR) significantly depend on the ambient temperatures. The comparison of temperatures inside cirrus at different latitudes shows no significant difference. Therefore, the detected difference in cirrus properties is likely not caused by temperatures.